# Maternally transmitted anti-measles antibodies, and susceptibility to disease among infants in Chandigarh, India: A prospective birth cohort study

Joseph L. Mathew[1]*, Abram L. Wagner[2], Radha Kanta Ratho[3], Pooja N. Patel[2], Vanita Suri[4], Bhavneet Bharti[1], Bradley F. Carlson[2], Sourabh Dutta[1], Mini P. Singh[3], Matthew L. Boulton[2,5]

1 Department of Pediatrics, Advanced Pediatrics Center, PGIMER, Chandigarh, India, 2 Department of Epidemiology, School of Public Health, University of Michigan, Ann Arbor, Michigan, United States of America, 3 Department of Virology, PGIMER, Chandigarh, India, 4 Department of Obstetrics and Gynecology, PGIMER, Chandigarh, India, 5 Department of Internal Medicine, Division of Infectious Disease, University of Michigan Medical School, Ann Arbor, Michigan, United States of America

* dr.joseph.l.mathew@gmail.com

**Data Availability Statement:** All relevant data are within the paper and its Supporting Information

## Abstract

Prior to the age of measles vaccination, infants are believed to be protected against measles by passively transferred maternal antibodies. However, the quantity and quality of such protection have not been well established in the Indian setting. We undertook this study to characterize the transfer and decline in maternal anti-measles antibodies among infants, and determine their susceptibility to measles. In this population-based, birth-cohort study, we enrolled pregnant women and their newborn infants, from a catchment area of 30 Anganwadis in Chandigarh, India. We collected maternal blood at delivery, and infant blood samples at birth, and 3, 6, and 9 months of age. Anti-measles IgG antibodies were measured using quantitative ELISA. We assessed antibody decline using log-linear models. In total, 428 mother-infant dyads were enrolled, and data from 413 dyads were analyzed. At birth, 91.5% (95% CI: 88.8, 94.2) of infants had protective antibody levels, which declined to 26.3% (95% CI: 21.0%, 31.9) at 3 months, 3.4% (95% CI: 0.9, 5.9) at 6 months, and 2.1% (95% CI: 0.1, 4.1) at 9 months. Younger mothers transferred lower levels of antibodies to their infants. We concluded that the majority of infants are susceptible to measles as early as three months of age, much earlier than their eligibility to receive measles vaccination.

## Introduction

Measles is one of the most infectious diseases in humans. Prior to widespread vaccination, it was responsible for substantial morbidity and mortality. Despite the impressive decline in global incidence from 145 cases per million population, to just 18 per million, from the year 2000 to 2016, it increased by nearly ten-fold during a worldwide resurgence in 2019 [1]. Recent data suggest that the number of cases increased by 79 per cent during the first two months of

files. The data are also available online at: https://doi.org/10.6084/m9.figshare.22190215.v2.

**Funding:** MLB Grant Number: Nil Trehan Family Foundation, USA https://www.nriinternet.com/NRIentrepreneurs/USA/A_Z/T/Ranvir.Trehan/index.htm The funders did not play any role in the study design, data collection and analysis, decision to publish, or preparation of the manuscript?

**Competing interests:** The authors have declared that no competing interests exist.

2022, compared to the preceding year [2]. During the year 2022, India contributed the highest number of cases globally [3]. By November 2022, the World Health Organization recorded 12,773 cases in India, making it the largest outbreak of the year [4].

For several decades, measles immunization in India comprised a single dose of vaccine administered to infants at 9 months of age. This age was chosen to balance the need for early protection against measles disease, versus preventing potential interference in the immune response by maternally transmitted antibodies. More recently, an additional dose was added through the measles-mumps-rubella (MMR) vaccine administered at 15 months. The current vaccination policy for infants consists of a dose of measles-rubella (MR) vaccine administered at 9 months, followed by MMR vaccine at 12–15 months. More recently, nationwide supplementary immunization with the MR vaccine was launched for children 9 months through 14 years of age [5].

During the interval between birth and the eligibility for measles vaccination at nine months, infants are expected to be protected by maternal anti-measles immunoglobulin G (IgG) antibodies, transferred to the fetus during the last trimester of pregnancy [6]. The titer of measles IgG antibodies above a protective threshold shields infants against measles infection [7]. However, it is now evident that as many as 4–12% infants in India develop measles before the age of 9 months [8–10]. A study in Kerala (southern India), showed that among 43 children with measles, one-third acquired it prior to the age of vaccination [11]. Analysis of outbreaks in urban slums in Delhi also reported significant numbers of cases among infants younger than 9 months, with the youngest case being just one month old [12]. During other outbreaks, cases were identified among infants as young as 3–6 months in northern as well as southern India [13–15]. Likewise, an analysis of measles cases among infants younger than 1 year old, showed that 62.5% were younger than 9 months old [16]. During an orphanage outbreak in Pune, five of six cases were younger than nine months; four were younger than 4 months old [17].

Such early infection could have developed because maternally transmitted antibodies wane at a faster rate in infants than previously estimated, which in turn could be due to lower antibody titers in mothers having immunity to measles through their own vaccination, rather than natural infection [18,19]. A cross-sectional study of infants identified sero-protection in only 4.7% and 2.7% of infants at six and nine months of age respectively [20]. Two small longitudinal studies in Chandigarh also confirmed that seroprotection in infants, from maternally transferred antibodies waned faster than expected. The first was conducted in 2005 (about 20 years after initiation of universal measles immunization) in 61 infants. The proportion of susceptible infants at birth, 3 months, 6 months and 9 months of age was 0%, 11.5%, 72%, and 94% respectively [21]. The second study conducted in 2015 (30 years after the initiation of universal measles vaccination) among 200 infants, showed the proportion of unprotected infants at birth, 3 months, 6 months and 9 months of age was 6%, 23%, 100%, and 100% respectively [22], suggesting declining protection during the intervening decade. Similarly in Tianjin, China, most infants were unprotected by 3 months of age [23]. These data suggest an epidemiological shift in India and other locations wherein infants are susceptible to measles well before the age of vaccination at 9 months.

Such observations argue for adequately powered longitudinal studies to understand the presence of maternally transmitted anti-measles antibodies, and susceptibility to disease among infants (until they reach the age when they are eligible for measles vaccination). This study was undertaken to address this knowledge gap, by measuring the levels of anti-measles antibodies in a cohort of infants at birth, correlating it with the level of antibodies in their mothers, and measuring the pattern of antibody decline at 3-month intervals, until the age of measles vaccination, i.e. 9 months. We also explored antibody patterns by sociodemographic status (caste, religion, income, education) under a 'social determinants of health' framework in

order to ascertain whether any groups could be affected (than others) by future measles outbreaks.

## Methods

This population-based study was conducted during 2016–17, in Chandigarh, a city and Union Territory (UT) of India, that serves as the capital of two northern Indian states- Punjab and Haryana. Its population of over 1 million residents [24] resides in villages, urban areas, and resettlement colonies (previously referred to as "urban slums"). Chandigarh has 510 units referred to as 'Anganwadis' [25], which administer the Governmental Integrated Child Development Services Scheme (ICDS), to address childhood malnutrition, infant vaccination, and antenatal registration of pregnant women. Each Anganwadi serves a population of about 2000 people. They are located across rural areas, urban areas, and resettlement colonies of Chandigarh. Prior to initiating the study, the 510 Anganwadis were listed, and a total of 30 were selected through a random selection process, and similarly 15 back-up Anganwadis were selected as reserve. The 30 shortllisted Anganwadis and 15 reserve Anganwadis were selected in proportion to the population residing in the rural areas, urban areas, and resettlement colonies. This ensured that the population in this study represented the population of Chandigarh.

### Study population

A total of 1483 pregnant women residing within the selected Anganwadis were approached during the third trimester, to explain (in the local vernacular) about the study, and its procedures. Those who expressed interest in participating were offered a detailed Participant Information Sheet (PIS) and Informed Consent Form (ICF). These were prepared as per the institutional rules, in three languages (English, Hindi, and Punjabi). Those who were willing to participate in the study were enrolled at the time of delivery, and their infants were enrolled immediately after delivery, with written, informed consent. Mothers were excluded if they had a health condition that hindered normal daily activities; or had an acute febrile illness, hemophilia, or any other blood disorder that could lead to excessive bleeding. Infants were excluded if they required resuscitation at birth or had any life-threatening congenital anomaly. The mother-infant dyads comprised the study population. The infants were followed longitudinally at 3, 6, and 9 months of age.

### Sample size

This study was designed to identify the decline in antibody titers with increasing age of the infants. A previous pilot study conducted in our institution showed that the proportion of susceptible (i.e. unprotected) infants was 0%, 11.5%, 72%, and 94% at birth, 3 months, 6 months and 9 months of age, respectively [21]. Based on this data, we wanted to choose an adequate sample size to describe changes in antibody titers across the nine months from 94% to 72% to 12% to 0%, and significantly distinguish these proportions. A sample size of 200 provided that level of precision, at an alpha of 0.05 and a power of 80%. It provided a margin of error of 4.6% for seropositivity values >90% and <10%, and 6.6% for values around 70%. This margin of error is adequate for our aim. Assuming approximately 50% attrition among infants over the duration of 9 months, we sought to enroll 400 mother-infant dyads. Since the estimation of antibody titers across sociodemographic groups was exploratory, we did not conduct separate sample size calculations for this.

## Case Record Form

We used a Case Record Form (CRF) to collect data of mothers (age, parity, educational status, income, caste, religion, residence, vaccination records, prior measles infection) and infants (birth date, gestation, weight). During the follow-up, the CRF included questions about symptoms and signs conforming to the clinical definition of measles infection in the infant and/or family members; blood sampling date, and antibody levels. The CRF was prepared in English, and translated to Hindi and Punjabi. It was pilot-tested for ease of use, consistency, and ability to capture the required data. Thereafter, it was finalized and deployed. In the field, each CRF required approximately 10 minutes time for completion.

## Study procedures

At enrollment, maternal measles history and vaccination history were documented, either by examination of records (where available), or recall. A 2mL blood specimen was collected by venipuncture from the mother just prior to delivery, and a 1mL sample of cord blood was collected from the infant at birth. During the follow-up visits at 3, 6, and 9 months of age, study personnel inquired about symptoms/signs of measles disease in the infant, or family members; breastfeeding and feeding practices; and infant anthropomorphic measurements. Three-month intervals were chosen to optimize the number of samples per infant, without over-burdening the infants or their families. At each follow-up visit, 1mL blood specimen was collected by venipuncture from infants. All blood sampling was performed by trained nurses who were hired for the study.

Blood samples were centrifuged to separate serum, which was aliquoted and stored at -20˚C for analysis in batches. Anti-measles IgG antibodies were measured in serum using an enzyme-linked immunosorbent assay (ELISA) kit (IMMUNOLAB Measles IgG Antibody ELISA Test Kit, IMMUNOLAB GmbH, Kassel, Germany). According to the manufacturer's instructions, titers >12 IU/mL were defined as positive/protective, 8 to 12 IU/mL as borderline/indeterminate, and <8 IU/mL as negative. Serum specimens with levels below the protective IgG antibody threshold were labelled as 'susceptible' to measles. Twenty percent of the stored specimens (selected randomly) were re-analyzed to evaluate variation in antibody levels, thereby ensuring quality control in the laboratory procedures. In addition, any specimens showing extremes of antibody levels were also re-analyzed twice. Laboratory testing was performed by a trained laboratory technician in accordance with the National Accreditation Board for Testing and Calibration Laboratories (NABL) Standards, ISO 17025 for public health laboratories [26], as well as the World Health Organization Manual for the laboratory diagnosis of measles and rubella, 2nd edition (WHO/IVB/07.01) [27].

## Outcome measures and derived variables

The primary outcome analyzed was the infant anti-measles IgG antibody titer and change in titer levels over time. The continuous antibody titer value was categorized into negative, borderline, and positive.

The covariates analyzed included infant data (age, sex, breastfeeding status); maternal data (age at delivery, parity, prior measles disease, measles vaccination, education); and family data (socioeconomic status, religion, caste). Mother's history of vaccination and natural measles infection were categorized into the following: neither (no vaccination, no history of measles); unknown for both; vaccination but no measles; and measles regardless of vaccination status. Parity greater than 2 was combined into one category. For religion, we collapsed together the Muslim and Christian categories; and for caste, the Scheduled Caste (SC) and Scheduled Tribe (ST) categories, due to low cell counts.

## Statistical analysis

We examined the proportion of infants who fell into the negative, borderline, and positive categories of anti-measles IgG antibody status at 0, 3, 6, and 9 months. The mean antibody titer was also calculated at each of these time points.

To characterize the decline in maternal antibodies among infants, descriptive statistics were first calculated to illustrate distributions of potential predictors. In a bivariate analysis, the geometric mean titer of anti-measles IgG at birth is displayed across age and different sociodemographic categories.

A multivariable log-linear model was constructed to estimate the association between infants' age and antibody titer. This model also included sociodemographic variables as an exploratory analysis. An autoregressive covariance structure was used to account for correlation between an infant's visits, with increasing variability across farther apart months. In order to obtain exp(β) and 95% confidence intervals (CIs) beta coefficients and lower/upper limits were exponentiated. Predictor variables were also tested for collinearity to ensure the absence of strong linear association. The model included the following predictors: infant's age, sex, breastfeeding status, family's income, religion, caste, mother's age, parity, education, and mother's disease/vaccination status. All predictors, except mother's age, were entered as categorical. Mother's age was continuous and not log transformed.

For an additional sensitivity analysis, we adjusted for the loss to follow up. The enrolled sample declined to 253 at 3 months, 204 at 6 months, and 181 at 9 months. We have described the possible reasons for the loss to follow up in a separate manuscript [15]. In our sensitivity analysis, we adjusted for the inverse probability of infants staying in the study. We calculated these weights based on the same set of covariates used in the final multivariable logistic regression model (infant's age, sex, breastfeeding status, family's income, religion, caste, mother's age, parity, education, and mother's disease/vaccination status).

All analyses were conducted using SAS version 9.4 (SAS Institute, Cary, NC, USA).

## Ethical approval

This study was approved by the Institutional Ethics Committee of PGIMER Chandigarh (PGI/IEC/2015/13630, the University of Michigan Health Sciences and Behavioral Sciences Institutional Review Board (#HUM00104905); and the Health Ministry Screening Committee, Ministry of Health and Family Welfare, Government of India. All participants were enrolled with written informed consent provided by self (for mothers) or either parent (for infants).

## Results

A total of 428 mother-infant dyads were enrolled in the study. However, 15 were excluded (7 infants did not meet the study criteria; 7 infants received a measles containing vaccine (MCV) during the study period; and 1 infant developed measles disease during the course of the study). Thus, a total of 413 mother-infant dyads were included in the analysis. Of the 413 infants, six visits for four infants were excluded due to missing antibody titer values or interview data. Table 1 shows the demographic characteristics of 413 infants and mothers at enrollment, and the infants' geometric mean titers at birth. In this study, there were seven women who gave birth to twins. We included both pairs of twins in all seven cases. However, the sample is too small to conduct specific sub-analyses.

The geometric mean anti-measles IgG antibody titer in the infants was 64 IU/mL (95% CI: 58, 71) at birth (n = 413), 8 IU/mL (95% CI: 7, 9) at 3 months (n = 253), 2 IU/mL (95% CI: 2, 3) at 6 months (n = 204), and 1 IU/mL (95% CI: 1, 2) at 9 months of age (n = 181) (Fig 1). The proportion of infants with protective antibody levels was 91.5% at birth (95% CI: 88.8, 94.2),

**Table 1. Baseline demographic characteristics of 413 infant-mother dyads at the time of birth.**

| Characteristic | Number | Percentage | Anti-measles IgG at birth | |
| --- | --- | --- | --- | --- |
| | | | Geometric mean titer | 95% CI |
| **Infant Sex** | | | | |
| Female | 211 | 51.1 | 67 | 58–77 |
| Male | 202 | 48.9 | 62 | 53–71 |
| **Family's monthly income** | | | | |
| < Rs 10,000 per month | 186 | 45.0 | 67 | 58–77 |
| Rs 10,000–25,000 per month | 121 | 29.3 | 64 | 53–78 |
| Rs 25,000–50,000 per month | 82 | 19.9 | 63 | 50–79 |
| > Rs 50,000 per month | 24 | 5.8 | 50 | 28–89 |
| **Family religion** | | | | |
| Hindu | 340 | 82.3 | 66 | 59–74 |
| Sikh | 55 | 13.3 | 54 | 40–73 |
| Muslim/Christian | 18 | 4.4 | 67 | 43–104 |
| **Family Caste** | | | | |
| General | 274 | 66.3 | 64 | 56–72 |
| Other Backward Caste | 54 | 13.1 | 66 | 49–90 |
| SC/ST | 85 | 20.6 | 65 | 52–81 |
| **Mother's Age (Mean, SD)** | 27.6 (4.6) | | | |
| 18–24 years | 105 | 25.4 | 63 | 50–78 |
| 25–29 years | 181 | 43.8 | 61 | 52–72 |
| ≥30 years | 127 | 30.8 | 70 | 60–83 |
| **Parity** | | | | |
| 0 | 264 | 63.9 | 64 | 56–73 |
| 1 | 118 | 28.6 | 63 | 53–75 |
| ≥2 | 31 | 7.5 | 69 | 52–91 |
| **Mother's Education** | | | | |
| None | 16 | 3.9 | 75 | 44–127 |
| Primary (≤5 years of schooling) | 16 | 3.9 | 77 | 47–127 |
| Middle (<8 years of schooling) | 34 | 8.2 | 58 | 41–83 |
| Higher secondary (<12 years of schooling) | 115 | 27.9 | 64 | 53–76 |
| College/University or higher | 232 | 56.2 | 64 | 56–74 |
| **Mother's Measles Disease/Vaccination Status** | | | | |
| No vaccination, no disease | 217 | 52.5 | 67 | 58–76 |
| Unknown/missing for both | 140 | 33.9 | 58 | 49–70 |
| Vaccination, no disease | 36 | 8.7 | 57 | 37–89 |
| Disease, regardless of vaccination | 20 | 4.8 | 108 | 76–152 |

26.9% at 3 months (95% CI: 21.0, 31.9), 3.4% at 6 months (95% CI: 0.9, 5.9), and 2.1% at 9 months (95% CI: 0.1, 4.1) (Fig 1). Geometric mean titers across different sociodemographic groups are shown in Table 1.

Table 2 shows results from the multivariable log-linear model with antibody titer as a continuous, log-transformed outcome.

Compared to infant antibody titers at birth, the titers were 0.12 times as high (95% CI: 0.06, 0.24) at 3 months of age, 0.03 times as high (95% CI: 0.02, 0.07) at 6 months of age, and 0.02 times as high (95% CI: 0.01, 0.04) at 9 months of age. For every one-year increase in maternal age at delivery, infant antibody titers were 1.03 times higher (95% CI: 1.01, 1.05). In a sensitivity analysis that accounted for differential loss to follow up across different demographic

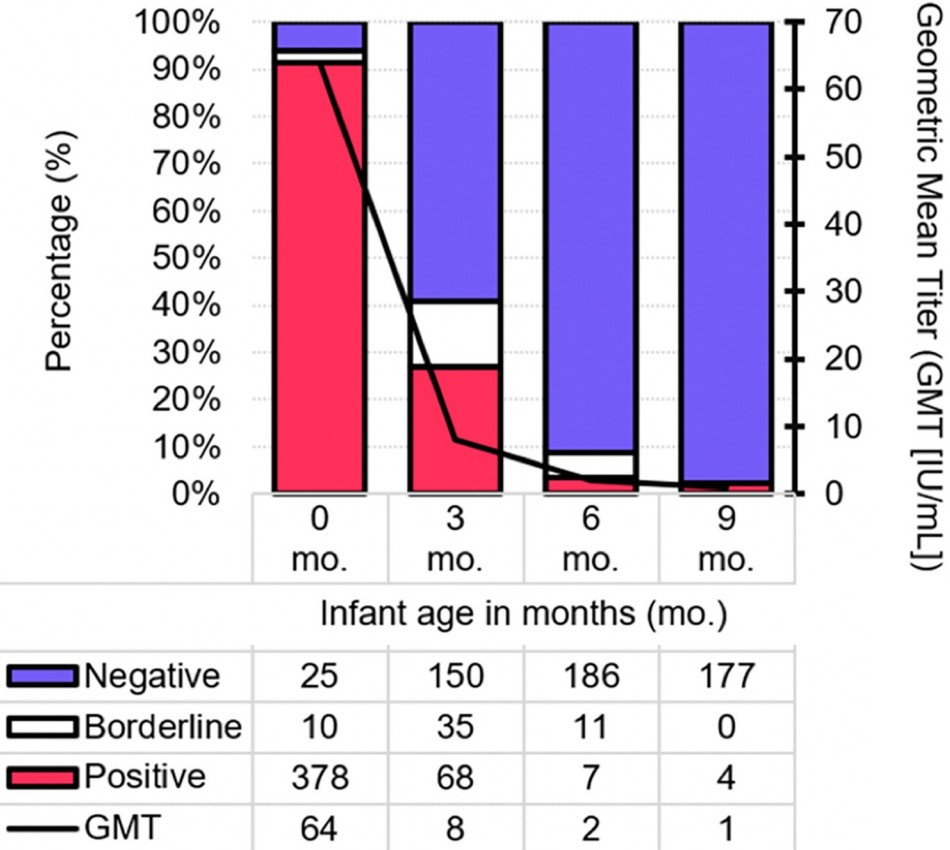

**Fig 1. Geometric mean antibody titers in infants from birth to 9 months, and proportions with protective levels.**

groups, these variables remained significant. Infant sex, breastfeeding status, maternal education level, measles disease and vaccination status; and family SES, religion, and caste, were not related with infants' antibody titer at any age.

## Discussion

In this longitudinal birth-cohort study of infants, we found a substantial decline in maternal-derived, anti-measles antibody titers in infants; most had lost protection by 3 months of age. We also identified that younger mothers (who probably have vaccine-derived measles immunity), were more likely to have unprotected infants.

This is not the first study highlighting the problem of infants remaining unprotected from measles, prior to the age of vaccination. Besides our own studies highlighted above [21,22], a study in 250 infants showed that only 25 (10%) had IgG antibodies just prior to vaccination [28]. Similarly, another cross-sectional study of 120 infants showed that 85% lacked antibodies prior to vaccination at the age of 9–10 months [29].

Similar experiences have been reported in other countries also. Data from Israel (where measles vaccine is administered at 1 year of age) showed that less than half of the infants were protected at 6 months of age, and just 4% were protected between 6–11 months of age [30]. A study in Turkey examined infants' antibody levels at one and six months of age. While 96% of one-month old infants were seropositive, only 25% remained so at 6 months of age, confirming earlier disappearance of maternal antibodies [31].

**Table 2. Results from multivariable log-linear model predicting log-transformed infant antibody titer using infant and mother characteristics (n = 413 dyads).**

| Category | Original model | | Weighted for loss to follow-up | |
|---|---|---|---|---|
| | exp(β) (95% CI) | P-value | exp(β) (95% CI) | P-value |
| **Infant's age in months** | | < .0001 | | < .0001 |
| 0 | ref | | ref | |
| 3 | 0.12 (0.06, 0.24) | | 0.09 (0.06, 0.15) | |
| 6 | 0.03 (0.02, 0.07) | | 0.03 (0.02, 0.05) | |
| 9 | 0.02 (0.01, 0.04) | | 0.02 (0.01, 0.02) | |
| **Infant's sex** | | 0.3362 | | 0.1149 |
| Male | ref | | ref | |
| Female | 1.08 (0.92, 1.26) | | 0.75 (0.55, 1.02) | |
| **Breastfed** | | 0.9830 | | 0.4911 |
| No | ref | | ref | |
| Yes | 1.01 (0.48, 2.11) | | 1.16 (0.76, 1.77) | |
| **Family's monthly income** | | 0.4349 | | 0.0429 |
| Rs <10,000 per month | ref | | ref | |
| Rs 10,000–25,000 per month | 1.03 (0.84, 1.26) | | 0.67 (0.43, 1.05) | |
| Rs 25,000–50,000 per month | 0.88 (0.69, 1.11) | | 0.20 (0.11, 0.37) | |
| Rs >50,000 per month | 0.79 (0.51, 1.24) | | 0.35 (0.17, 0.71) | |
| **Family religion** | | 0.2076 | | 0.3380 |
| Hindu | ref | | ref | |
| Sikh | 0.82 (0.65, 1.03) | | 0.18 (0.10, 0.33) | |
| Muslim/Christian | 1.10 (0.80, 1.50) | | 1.04 (0.58, 1.89) | |
| **Family caste** | | 0.8566 | | 0.1912 |
| Other | ref | | ref | |
| Other Backwards Caste | 1.01 (0.79, 1.28) | | 1.06 (0.72, 1.56) | |
| SC/ST | 0.95 (0.78, 1.16) | | 0.73 (0.51, 1.03) | |
| **Mother's Age** | 1.03 (1.01, 1.05) | 0.0068 | 1.13 (1.08, 1.17) | < .0001 |
| **Parity** | | 0.6763 | | 0.4816 |
| 0 | ref | | ref | |
| 1 | 0.94 (0.78, 1.12) | | 1.14 (0.84, 1.54) | |
| ≥2 | 0.90 (0.68, 1.19) | | 0.78 (0.41, 1.49) | |
| **Mother's Education** | | 0.7395 | | 0.1121 |
| None/Other | ref | | ref | |
| Primary (≤5 years of schooling) | 1.10 (0.67, 1.79) | | 1.02 (0.34, 3.05) | |
| Middle (<8 years of schooling) | 0.84 (0.54, 1.29) | | 0.39 (0.18, 0.84) | |
| Higher secondary (<12 years of schooling) | 0.93 (0.62, 1.39) | | 0.59 (0.29, 1.18) | |
| College/University or higher | 0.89 (0.59, 1.34) | | 1.07 (0.59, 1.94) | |
| **Mother's Measles Disease/Vaccination Status** | | 0.1159 | | 0.1182 |
| No vaccination, no disease | ref | | ref | |
| Unknown/missing for both | 0.90 (0.75, 1.08) | | 0.28 (0.15, 0.52) | |
| Vaccination, no disease | 0.85 (0.61, 1.18) | | 0.56 (0.35, 0.91) | |
| Disease, regardless of vaccination | 1.37 (1.00, 1.88) | | 2.03 (0.83, 4.98) | |

Our observations have several implications for public health and policy-making. First, it questions the current practice of vaccinating infants at 9 months of age in India. This age was chosen in most developing countries to reduce the chance of infection in early infancy. Although earlier vaccination seems an attractive option to reduce the risk of leaving young infants unprotected, the immature infant immune system, and the potential interference by even low titers of circulating maternal antibodies [6,18,23,32] necessitate caution.

Second, what could be the optimal timing of infant vaccination? Some investigators did attempt to examine the efficacy of earlier measles vaccination in India. One such pioneering study reported seroconversion in 74% infants vaccinated at 6 months of age, compared to 100% at 13–15 months, although the difference was not statistically significant [33]. Another study compared the long-term protection of earlier vaccination. Among infants vaccinated at 6–8 months of age, only 49% had protective levels of antibodies when they were 16–44 months old, compared to 67% children vaccinated after 8 months [34]. In contrast, a pilot study comparing vaccination at 6–8 months versus 9–11 months, showed no statistically significant difference in the risk of developing measles, or mortality due to measles, with either regimen [35]. Studies such as these led the WHO to endorse measles vaccine at 9 months of age in developing countries [36]. However, now that one or more additional dose(s) of measles-containing vaccine is/are universally recommended [37], it may be feasible to anticipate the age of infant vaccination. There is also emerging data to suggest that it may be cost-effective to vaccinate individuals at an earlier age [38]. However, determining the optimal age of infant vaccination would require well-designed randomized controlled trials, with adequate follow-up and meticulous monitoring to ensure that infants are adequately protected against measles.

Third, the potential mechanism for infants becoming susceptible much before the age of vaccination, needs careful consideration. Data from the USA where measles vaccination was introduced several decades back, is illustrative. Among US-born women, antibody titers decreased with increasing birth year. In contrast, there was no such difference among women born outside the US. Further, the infants of younger US-born women were less likely to have measles antibodies at 6, 9, or 12 months of age (i.e. prior to their eligibility for vaccination), compared to infants of older women [39]. This suggests that vaccine-induced immunity is less robust than that induced by natural infection, and it impacts not only the vaccinated women, but the next generation as well. However, in societies where measles has not been eliminated, the impact of subclinical and clinical infections in the community, on the antibody levels of mothers (and their consequent impact on infants) is unclear. We did not find a significant correlation between mothers' disease or vaccination status, and their infant's antibody titers, probably because our study is underpowered to detect this difference.

Fourth, these data raise the possibility that higher antibody titers in pregnant women, could result in transmission of enough antibodies to infants, to protect them until the age of vaccination. In such a situation, could vaccination of adolescent girls, or women of child-bearing age, be leveraged for infant protection? Vaccination during pregnancy may be an alternate option. However, a previous study from China reported that even among the infants whose mothers had the highest antibody titers, a large proportion were seronegative by 3 months of age [23].

## Strengths and limitations

A strength of this study was the use of a population-based, longitudinal birth-cohort design to assess decline in measles antibody titers in young infants. We achieved the pre-determined sample size of 200 mother-infant dyads, except for the last evaluation at 9 months. We also collected data on several additional variables that impact infant vaccination and susceptibility to disease, but none showed statistically significant impacts on anti-measles antibody levels.

However, we acknowledge several limitations. There was substantial loss to follow up, which was somewhat expected. We had previously documented that this could have been due, in part, to individuals switching cell phone service providers [40]. However, sensitivity analysis accounting for demographic correlates of loss to follow up did not reveal any substantial differences. We did not have the resources to perform analysis of serum for anti-measles IgM antibodies (to document acute subclinical infection) at each visit.

## Conclusion

This longitudinal birth-cohort study identified that the majority of infants were susceptible to measles, well before the age of vaccination. This suggests the need to protect them either through anticipated vaccination schedules, or enhancing the antibody levels in pregnant women. Meanwhile, we recommend strengthening of surveillance to determine if these susceptible infants develop clinical or subclinical measle.

## Supporting information

**S1 File.**
(CSV)

## Acknowledgments

We would like to thank the research staff for their invaluable role in survey administration and data collection, and Anganwadi workers/community leaders who facilitated participant selection. Preliminary data from this study was presented at a conference but no results from the final data set have been previously presented.

## Author Contributions

**Conceptualization:** Joseph L. Mathew, Abram L. Wagner, Matthew L. Boulton.

**Data curation:** Joseph L. Mathew, Abram L. Wagner, Radha Kanta Ratho, Pooja N. Patel, Bradley F. Carlson, Sourabh Dutta, Mini P. Singh, Matthew L. Boulton.

**Formal analysis:** Joseph L. Mathew, Abram L. Wagner, Radha Kanta Ratho, Pooja N. Patel, Vanita Suri, Bradley F. Carlson, Matthew L. Boulton.

**Funding acquisition:** Matthew L. Boulton.

**Investigation:** Joseph L. Mathew, Radha Kanta Ratho, Vanita Suri, Bhavneet Bharti, Bradley F. Carlson, Sourabh Dutta, Mini P. Singh.

**Methodology:** Joseph L. Mathew, Abram L. Wagner, Radha Kanta Ratho, Mini P. Singh, Matthew L. Boulton.

**Project administration:** Joseph L. Mathew, Abram L. Wagner, Vanita Suri, Bhavneet Bharti, Bradley F. Carlson, Matthew L. Boulton.

**Resources:** Joseph L. Mathew.

**Software:** Pooja N. Patel, Bradley F. Carlson, Matthew L. Boulton.

**Supervision:** Joseph L. Mathew, Vanita Suri, Bhavneet Bharti, Matthew L. Boulton.

**Validation:** Joseph L. Mathew, Abram L. Wagner, Radha Kanta Ratho, Pooja N. Patel, Bhavneet Bharti, Bradley F. Carlson, Sourabh Dutta, Mini P. Singh.

**Visualization:** Joseph L. Mathew, Abram L. Wagner, Radha Kanta Ratho, Pooja N. Patel, Vanita Suri, Bhavneet Bharti, Bradley F. Carlson, Sourabh Dutta, Mini P. Singh, Matthew L. Boulton.

**Writing – original draft:** Joseph L. Mathew, Abram L. Wagner, Pooja N. Patel.

**Writing – review & editing:** Joseph L. Mathew, Abram L. Wagner, Radha Kanta Ratho, Pooja N. Patel, Vanita Suri, Bhavneet Bharti, Bradley F. Carlson, Sourabh Dutta, Mini P. Singh, Matthew L. Boulton.

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
