## [Decision Letter · Decision Letter 0]

8 Jan 2023

PONE-D-22-30962Measles antibodies and susceptibility to disease among infants in Chandigarh, India:  A prospective birth cohort studyPLOS ONE

Dear Dr. Mathew,

Thank you for submitting your manuscript to PLOS ONE. After careful consideration, we feel that it has merit but does not fully meet PLOS ONE’s publication criteria as it currently stands. Therefore, we invite you to submit a revised version of the manuscript that addresses the points raised during the review process.

We look forward to receiving your revised manuscript.

Kind regards,

Ghada Abdrabo Abdellatif Elshaarawy, M.D

Academic Editor

PLOS ONE

Journal Requirements:

Additional Editor Comments:

1) ABSTRACT:

A brief background, plus the aim, should be added.

2) INTRODUCTION:

Newest Global/ Regional/ India/ Chandigarh prevalence of measles cases in children younger than 9 months should be stated.

Risk factors associated with Measles antibodies among infants in Chandigarh, India should be clearly stated.

3) METHODS:

There was no clear mention of the questionnaire used. What literature was reviewed to develop the questionnaire? The authors do not provide any references. Was it piloted to assess its internal consistencies? Was it validated? Additional explanations about the validity and reliability of the questionnaire should be added - explained step by step. (Content and structure validity). How long did it take to complete each questionnaire?

Tests of significance should be added.

4) RESULTS:

Table 2 is not clear; it needs more explanation.

5) DISCUSSION:

Discuss by using the scientific reasoning the anti-measles antibody titers among children younger than 9 months in other developing and developed countries with similar context. The manuscript could be greatly strengthened if the authors could compare the findings of the study with other findings and state the reasons for the strengths and weaknesses in each section.

Reviewers' comments:

Reviewer's Responses to Questions

**Comments to the Author**

1. Is the manuscript technically sound, and do the data support the conclusions?

Reviewer #1: Yes

Reviewer #2: Yes

Reviewer #3: Yes

2. Has the statistical analysis been performed appropriately and rigorously? 

Reviewer #1: Yes

Reviewer #2: Yes

Reviewer #3: Yes

3. Have the authors made all data underlying the findings in their manuscript fully available?

Reviewer #1: Yes

Reviewer #2: No

Reviewer #3: Yes

4. Is the manuscript presented in an intelligible fashion and written in standard English?

Reviewer #1: Yes

Reviewer #2: Yes

Reviewer #3: Yes

5. Review Comments to the Author

Reviewer #1: Dear PLOS ONE team of Editorials, thank you for the chance given to me to review “Measles antibodies and susceptibility to disease among infants in Chandigarh, India: A prospective birth cohort study” with manuscript Number PONE-D-22-30962. The study will escalate our knowledge and will update on the antibody of Measles and the susceptibility and its consequent protective rate as well. The following are my comments:

1. We know that the measles vaccine is protective for at least 9 months transferring from mother to child in LMIC and 12 months in high income countries taking in to account e.g., the absence of measles epidemics. Hence, is the study area/ Chandigarh, India/ not measles epidemic affected or epidemic prone area? Why at third, sixth month? You had measured the maternal IgG antibody once. Who measured? How many? What is the qualification to collect the sample? Have you tested the inter-personal reliability? Have you assured quality of both the maternal and newborn biological sample? Do you have SOP? Is that national? etc

2. Why you choose to conduct there? What is unique about the study area? The pre-pregnancy and pregnancy characteristics of the respondents and additional characters should be described.

3. It is longitudinal study with repeated measurements why you prefer longitudinal analysis over others?

4. The outcome variable is the measles susceptibility and protection status of the maternal transferred anti-body against measles? Re-write the title again.

5. The selection of the women and the newborns should be revisited. E.g., If mothers give town newborn, how did you manage?

6. What is the prevalence and rate Measles in the study area?

7. What is the ICC level and why you prefer auto-regressive covariance over others?

8. The recommendation should be drawn from your findings. E.g., the recommendation should be RCT of the Measles anti body study transfer and the susceptibility and protection of Newborns from measles infection.

9. Tables are not self-explanatory, sentences are incomplete, revisit the language and the sentence again.

10. Check and consult for the statistics.

Reviewer #2: The topic of interest is relevant and appropriate. However,

1. Why was the study published after a gap of 4 yrs?

2. What is the role of maternal educational status on the antibody titres?

3. What is the incidence of measles cases in infants under 9 months?

Reviewer #3: There is the need for the authors to show how they arrived at the sample size in a more explicit manner. The relevance of the essence of collecting data on religion and cast has not been optimally utilised to arrive at a conclusion on whether these have any influence on the results obtained. There is the need to explain more about this in the discussion section. These could be related to the attitude and practice of the people towards vaccination or be responsible for susceptibility to infection.

Was there any difference in the data obtained in rural, urban and resettlement colonies of Chandigarh? This was not reflected in any way in the discussion. What was the relevance of the anthropomorphic measurements?

6. PLOS authors have the option to publish the peer review history of their article (what does this mean?). If published, this will include your full peer review and any attached files.

Reviewer #1: No

Reviewer #2: No

Reviewer #3: No

---

## [Author Response · Author response to Decision Letter 0]

21 Feb 2023

Response to Editorial and Reviewers’ comments

We are grateful to the Editor and the Reviewers for their careful appraisal of our manuscript. We have considered each comment carefully and addressed each point. We believe that this has significantly enhanced the manuscript. 

Please find enclosed herewith:

1) A detailed point by point response to the comments and suggestions offered.

2) Revised manuscript with Track changes. All newly added text is also highlighted in blue color for easy identification. 

3) Revised manuscript (clean copy). 

We hope that the revised manuscript is acceptable to be considered for publication.

Thanking you,

Yours sincerely,

Joseph L. Mathew

20 Feb 2023

PlosOne REVIEWER COMMENTS

Dear Dr. Mathew,

Thank you for submitting your manuscript to PLOS ONE. After careful consideration, we feel that it has merit but does not fully meet PLOS ONE’s publication criteria as it currently stands. Therefore, we invite you to submit a revised version of the manuscript that addresses the points raised during the review process.

We look forward to receiving your revised manuscript.

Kind regards,

Ghada Abdrabo Abdellatif Elshaarawy, M.D

Academic Editor

PLOS ONE

Journal Requirements:

Comment

Response: Thank you for the advice. We have checked and confirm that the manuscript meets the requirements listed. 

Comment

Response: Thank you for the advice. The Title page has been included in the manuscript file.

Comment

Response: The datasets have been submitted along with the revised manuscript. 

Comment

Response: The datasets have been submitted along with the revised manuscript. 

Additional Editor Comments:

Comment

1) ABSTRACT:

A brief background, plus the aim, should be added.

Response: Thank you for the advice. This has been added.

Revised text:

Prior to the age of measles vaccination, infants are believed to be protected against measles by passively transferred maternal antibodies. However, the quantity and quality of such protection have not been well established. We undertook this study to characterize the transfer and decline in maternal anti-measles antibodies among infants, and determine their susceptibility to measles. In this population-based, birth-cohort study, we enrolled pregnant women and their newborn infants, from a catchment area of 30 anganwadis in Chandigarh, India. We collected maternal blood at delivery, and infant samples at birth, and 3, 6, and 9 months of age. Anti-measles IgG antibodies were measured using quantitative ELISA. We assessed antibody decline using log-linear models. In total, 413 mother-infant dyads were enrolled. At birth, 91.5% (95% CI: 88.8, 94.2) of infants had protective antibody levels, which declined to 26.3% (95% CI: 21.0%, 31.9) at 3 months, 3.4% (95% CI: 0.9, 5.9) at 6 months, and 2.1% (95% CI: 0.1, 4.1) at 9 months. Younger mothers transferred lower levels of antibodies to their infants. We concluded that the majority of infants are susceptible to measles as early as three months of age, suggesting the need to anticipate measles vaccination to protect them. 

Comment

2) INTRODUCTION:

Newest Global/ Regional/ India/ Chandigarh prevalence of measles cases in children younger than 9 months should be stated.

Risk factors associated with Measles antibodies among infants in Chandigarh, India should be clearly stated.

Response: We have revised the text extensively as suggested. Global and Indian data have been included. However, there is no previous study reporting measles prevalence and its risk factors in Chandigarh union territory. 

Revised text:

Despite the impressive decline in global incidence from 145 cases per million population, to just to 18, during 2000 to 2016, it increased by nearly ten-fold during a global resurgence in 2019 [1]. Recent data suggest that the number of worldwide cases increased by 79 per cent during the first two months of 2022, compared to the preceding year [2]. During 2022, India reportedly contributed the highest number of cases globally [3]. By November 2022, the World Health Organization recorded 12,773 cases, making it the largest outbreak of the year [4]. Despite the impressive decline in global incidence from 145 cases per million population, to just to 18, during 2000 to 2016, it increased by nearly ten-fold during a global resurgence in 2019 [1]. Recent data suggest that the number of worldwide cases increased by 79 per cent during the first two months of 2022, compared to the preceding year [2]. During 2022, India reportedly contributed the highest number of cases globally [3]. By November 2022, the World Health Organization recorded 12,773 cases, making it the largest outbreak of the year [4].

Comment

3) METHODS:

There was no clear mention of the questionnaire used. What literature was reviewed to develop the questionnaire? The authors do not provide any references. Was it piloted to assess its internal consistencies? Was it validated? Additional explanations about the validity and reliability of the questionnaire should be added - explained step by step. (Content and structure validity). How long did it take to complete each questionnaire?

Tests of significance should be added.

Response: 

We used a simple Case Record Form (CRF) for the enrolled pregnant women (mothers). The CRF comprised questions related to maternal demographics (age, parity, caste, religion, residence, availability of vaccination records, documented prior measles infection, undocumented measles infection). Infant data (date of birth, gestation, birth weight, etc.) were recorded from the hospital records available at birth. During the follow-up, the CRF included questions about symptoms and signs of measles infection in the infant and/or family members (cough, coryza, conjunctivitis, presence of a maculopapular rash, physician confirmation of measles). Data of blood sampling date, and antibody levels, were also recorded.

The CRF was prepared in English, and translated to the local vernacular (Hindi and Punjabi). It was pilot tested for ease of use, consistency, and ability to capture the required data. Thereafter, it was finalized and deployed. In the field, each CRF required approximately 10 minutes time for completion. 

As the focus of this study is not a qualitative field survey (but rather a serial quantification of anti-measles antibody levels), these details were omitted for brevity. 

Revised text:

We used a Case Record Form (CRF) to collect data of mothers (age, parity, caste, religion, residence, vaccination records, prior measles infection) and infants (birth date, gestation, weight). During the follow-up, the CRF included questions about symptoms and signs conforming to the clinical definition of measles infection in the infant and/or family members; blood sampling date, and antibody levels. The CRF was prepared in English, and translated to Hindi and Punjabi. It was pilot-tested for ease of use, consistency, and ability to capture the required data. Thereafter, it was finalized and deployed. In the field, each CRF required approximately 10 minutes time for completion. 

Comment

4) RESULTS:

Table 2 is not clear; it needs more explanation.

Response: Table 2 presents two sets of analyses. First, a total of ten infant and maternal variables were examined to assess their impact on infants’ antibody levels. For each variable, one category was chosen as the reference (e.g. for infants’ sex, ‘male’ was chosen as the reference). This enabled calculation of exp(β) which is the factor by which the relative risk (of each category vs. the reference category) is multiplied, when the variable increases by one unit. 

To account for participant attrition over time, the analysis was repeated weighting for the loss to follow-up. This enabled us to assess the impact of participant attrition, on the results. Only two variables retained statistical significance across both analysis ,viz. infants’ age and maternal age. 

Revised text:

Table 2 shows results from the multivariable log-linear model with antibody titer as a continuous, log-transformed outcome. A total of ten infant and maternal variables were examined to assess their impact on infants’ antibody levels. For each variable, one category was chosen as the reference, enabling calculation of exp(β) (factor by which the relative risk of each category vs. the reference category, is multiplied, when the variable increases by one unit). To account for participant attrition over time, the analysis was repeated weighting for the loss to follow-up. Only two variables retained statistical significance across both analysis, viz. infants’ age and maternal age.

Comment

4) DISCUSSION:

Discuss by using the scientific reasoning the anti-measles antibody titers among children younger than 9 months in other developing and developed countries with similar context. The manuscript could be greatly strengthened if the authors could compare the findings of the study with other findings and state the reasons for the strengths and weaknesses in each section.

Response: Thank you for the advice. We have added the required text in the Introduction section, to justify the need for this study. 

Revised text:

New text added in the Introduction section:

Data from different parts of India suggest that 4-12% cases of measles occurred in infants younger than 9 months [8-10]. Analysis of outbreaks in urban slums in Delhi reported significant numbers of cases among infants younger than 9 months, with the youngest case being just one month old [11]. A study in Kerala (southern India), showed that among 43 children with measles, one-third acquired it prior to the age of vaccination [12]. During outbreaks, cases were identified among infants as young as 3-6 months in northern as well as southern India [13-15]. Likewise, an analysis of measles cases among infants younger than 1 year old, showed that 62.5% were younger than 9 months [16]. During an orphanage outbreak in Pune, five of six cases were younger than nine months; four were younger than 4 months old [17].

New text added in the Discussion section:

This is not the first study highlighting the problem of infants remaining unprotected from measles prior to the age of vaccination. Besides our own studies highlighted above [21,22], a study in 250 infants showed that only 25 (10%) had IgG antibodies just prior to vaccination [26]. Similarly, another cross-sectional study of 120 infants showed that 85% lacked antibodies prior to vaccination at the age of 9-10 months [27]. 

Similar experiences have been reported in other countries also. Data from Israel (where measles vaccine is first administered at 1 year of age) showed that less than half of the infants were protected at 6 months of age, and just 4% were protected between 6-11 months of age [28] A study in Turkey examined infants’ antibody levels at one and six months of age. While, 96% of one-month old infants were seropositive, only 25% remained so at 6 months of age, confirming earlier disappearance of maternal antibodies [29].

Unlike the previous cross-sectional studies, our findings are based on a longitudinal cohort of infants. Together, these observations have several implications for public health and policy-making. 

5) Reviewers' comments:

Reviewer's Responses to Questions

Comments to the Author

Comment

1. Is the manuscript technically sound, and do the data support the conclusions?

Reviewer #1: Yes

Reviewer #2: Yes

Reviewer #3: Yes

Response: Thank you.

Comment

2. Has the statistical analysis been performed appropriately and rigorously? 

Reviewer #1: Yes

Reviewer #2: Yes

Reviewer #3: Yes

Response: Thank you.

Comment

3. Have the authors made all data underlying the findings in their manuscript fully available?

Reviewer #1: Yes

Reviewer #2: No

Reviewer #3: Yes

Response: Thank you for the advice. The dataset has been uploaded.

Comment

4. Is the manuscript presented in an intelligible fashion and written in standard English?

Reviewer #1: Yes

Reviewer #2: Yes

Reviewer #3: Yes

Response: Thank you.

Comment

5. Review Comments to the Author

Reviewer #1: Dear PLOS ONE team of Editorials, thank you for the chance given to me to review “Measles antibodies and susceptibility to disease among infants in Chandigarh, India: A prospective birth cohort study” with manuscript Number PONE-D-22-30962. The study will escalate our knowledge and will update on the antibody of Measles and the susceptibility and its consequent protective rate as well. The following are my comments:

1. We know that the measles vaccine is protective for at least 9 months transferring from mother to child in LMIC and 12 months in high income countries taking in to account e.g., the absence of measles epidemics. Hence, is the study area/ Chandigarh, India/ not measles epidemic affected or epidemic prone area? Why at third, sixth month? You had measured the maternal IgG antibody once. Who measured? How many? What is the qualification to collect the sample? Have you tested the inter-personal reliability? Have you assured quality of both the maternal and newborn biological sample? Do you have SOP? Is that national? Etc

Response: Thank you.

1. Infants in LMICs were believed to be protected from measles till 6-9 months of age, by trans-placentally transferred maternal antibodies. This was based on limited cross-sectional, studies done in infants at various ages, in the pre-vaccination era. As measles was rampant in LMICs in the pre-vaccination era, most mothers had antibodies acquired through natural infection. In contrast, mothers in the current era (who were vaccinated when they were infants) have antibodies acquired through vaccination. Antibodies generated through natural infection are generally of higher titer, and last longer than vaccine-induced antibodies. In the current era, the pattern of antibody decline in infants has not been studied in a prospective cohort, although it is reasonable to expect that the levels would be lower and disappear faster (than among infants whose mothers developed antibodies through natural infection). The findings of this study support this viewpoint. 

The early disappearance of maternal antibodies makes infants susceptible to measles, a fact which is borne out by witnessing clinical measles among infants younger than 9 months. However, it may be insufficient to create an epidemic. 

2. India, like most developing countries has a fairly robust universal infant immunization program. Chandigarh union territory has one of the highest vaccination rates in the country. Previously outbreaks have been reported in Chandigarh, although not an epidemic. 

3. Intervals of 3 months were chosen for infant sampling, so that at least four samples could be obtained prior to the age of vaccination. Less frequent sampling would make it difficult to interpret results. More frequent sampling could provide rich data, but would be burdensome to infants and their family. 

4. Maternal and infant blood samples were collected by trained nurses who were recruited for the project work. All laboratory testing was performed at PGIMER by a trained laboratory technician recruited for the Project, and in accordance with the National Accreditation Board for Testing and Calibration Laboratories (NABL) Standards, ISO 17025 for public health laboratories, as well as the Manual for the laboratory diagnosis of measles and rubella, 2nd edition (WHO/IVB/07.01), WHO, Geneva, Switzerland. Quality system essential elements identified in CLSI GP26A3, that specifically apply to this project met these standards.

Revised text: The following sentences have been added (at different places) in the revised manuscript. 

All blood sampling was performed by trained nurses who were hired for the study.

Three-month intervals were chosen to optimize the number of samples per infant, without over-burdening the infants or their families

Laboratory testing was performed by a trained laboratory technician in accordance with the National Accreditation Board for Testing and Calibration Laboratories (NABL) Standards, ISO 17025 for public health laboratories, as well as the World Health Organization Manual for the laboratory diagnosis of measles and rubella, 2nd edition (WHO/IVB/07.01).

Comment

2. Why you choose to conduct there? What is unique about the study area? The pre-pregnancy and pregnancy characteristics of the respondents and additional characters should be described.

Response: Chandigarh union territory has a well-defined geographic area, with a well-documented population neatly categorised into urban, rural, and urban-resettlement colony (formerly called urban slum) areas. Additionally, Chandigarh has a total of 510 well-functioning anganwadi units, that cater to the basic health needs of a pre-specified population (in particular pregnant mothers and pre-school children). Third, Chandigarh has not been involved in large-scale intervention studies that could alter the basic epidemiology of infectious diseases. Last, Chandigarh has a fairly high coverage with routine childhood vaccines, and a robust system of recording infant vaccination. For these reasons, it is an ideal area to study the epidemiology of measles. 

Revised text: None

Comment

3. It is longitudinal study with repeated measurements why you prefer longitudinal analysis over others?

Response: It is well known that prospective, longitudinal, cohort studies are superior to cross-sectional studies at multiple time points. Performing cross-sectional studies of infants at different ages, would not permit a correlation between their maternal antibody levels at delivery, and the dynamics of waning of the maternally transferred antibodies. 

Revised text: None

Comment

4. The outcome variable is the measles susceptibility and protection status of the maternal transferred anti-body against measles? Re-write the title again.

Response: Thank you for the suggestion. The title has been revised as follows.

Revised text: 

Dynamics of maternally transmitted anti-measles antibodies and susceptibility to disease among infants in Chandigarh, India: A prospective birth cohort study.

Comment

5. The selection of the women and the newborns should be revisited. E.g., If mothers give town newborn, how did you manage?

Response: We respectfully state that this question is unclear. As stated in the Methods section:

Pregnant women residing within the 30 anganwadi clusters were approached during the third trimester of pregnancy, by research personnel to explain (in the local vernacular) about the study, its purpose and procedures. The pregnant women who were willing to participate in the study were enrolled at the time of delivery, and their infants were enrolled immediately after delivery. Mothers with a health condition that hindered normal daily activities; or who had acute febrile illness, hemophilia, or any other blood disorder that could lead to excessive bleeding, were excluded. Infants were excluded if they required resuscitation at birth or had any life-threatening congenital anomaly. The mother-infant dyads comprised the study population. Infants were followed longitudinally at 3, 6, 9, and 12 months of age. Chandigarh has a fixed number of institutions where pregnant women deliver babies. The PI’s institution caters to the bulk of the deliveries. 

Revised text: Please advise.

Comment

6. What is the prevalence and rate Measles in the study area?

Response: There is no robust, population data on the prevalence and rate of measles in the study area (Chandigarh) available in the public domain. However, the area has witnessed periodic outbreaks, three of which have been published (please see the citations below). Based on this, the Department of Virology at the study institution (PGIMER Chandigarh) was designated a nodal laboratory for measles outbreak response. 

Thakur JS, Ratho RK, Bhatia SP, Grover R, Issaivanan M, Ahmed B, Parmar V, Swami HM. Measles outbreak in a Periurban area of Chandigarh: need for improving vaccine coverage and strengthening surveillance. Indian J Pediatr. 2002 Jan;69(1):33-7.

Sharma MK, Bhatia V, Swami HM. Outbreak of measles amongst vaccinated children in a slum of Chandigarh. Indian J Med Sci. 2004 Feb;58(2):47-53

Ratho RK, Mishra B, Singh T, Rao P, Kumar R. Measles outbreak in a migrant population. Indian J Pediatr. 2005 Oct;72(10):893-4.

Revised text: Please advise if the above should be inserted into the manuscript. 

Comment

7. What is the ICC level and why you prefer auto-regressive covariance over others?

Response: In general, autoregressive covariance is appropriate when the correlation between responses decays over time, as in this study. An unstructured covariance is generally used when the variance is heterogenous. 

Revised text: Please advise if this needs to be inserted into the manuscript. 

Comment

8. The recommendation should be drawn from your findings. E.g., the recommendation should be RCT of the Measles anti body study transfer and the susceptibility and protection of Newborns from measles infection.

Response: Thank you for the advice. The Conclusion has been modified as shown below. 

Revised text: 

Discussion section:

However, determining the optimal age of infant vaccination would require well-designed randomized controlled trials, with adequate follow-up and meticulous monitoring to ensure that infants are adequately protected against measles.

Conclusion section:

This longitudinal birth-cohort study identified that the majority of infants were susceptible to measles, well before the age of vaccination. This suggests the need to protect them either through anticipated vaccination schedules, or enhancing the antibody levels in pregnant women. Meanwhile, we recommend strengthening of surveillance to determine if these susceptible infants develop clinical or subclinical measles.

Comment

9. Tables are not self-explanatory, sentences are incomplete, revisit the language and the sentence again.

Response: Thank you for the advice. The Tables and associated text have been revised. 

Revised text: 

Please see the revised tables and text.

Comment

10. Check and consult for the statistics.

Response: Thank you for the advice. The statistical analysis has been rechecked and confirmed. 

Comment

Reviewer #2: The topic of interest is relevant and appropriate. However,

1. Why was the study published after a gap of 4 yrs?

Response: We acknowledge a delay in manuscript preparation and submission. The investigators worked on other manuscripts related to the study in the interim. 

Comment

2. What is the role of maternal educational status on the antibody titres?

Response: In general, infant vaccination rates in developing countries (including India) are higher when mothers are better educated. Therefore, maternal education is always included as a variable in studies of infant immunization and immunity. 

Maternal education levels could indeed impact their antibody levels. Better education is an indirect marker of women empowerment, and societal structure. Better educated mothers tend to hail from families with better socio-economic backgrounds, and greater emphasis on woman independence. In such a situation, better educated mothers are more likely to have been vaccinated against measles when they were infants. This could account for relatively lower titres of antibodies in them, and hence lower levels in their infants. 

On the other hand, limited data suggest that the reverse is also possible. A case-control study in China identified that low education level of the primary caregiver was associated with a higher risk of measles among infants younger than 8 months [Tang ZZ, Xie YH, Jiraphongsa C, Liu XH, Li ZY, Chongsuvivatwong V. Risk factors for measles in children younger than age 8 months: A case-control study during an outbreak in Guangxi, China, 2013. Am J Infect Control. 2016 Apr 1;44(4):e51-8.].

However, as the study data did not conclusively prove that maternal education was inversely related to infant antibody levels, we did not highlight this point. 

Comment

3. What is the incidence of measles cases in infants under 9 months?

Response: There is limited robust data on the burden of measles cases among infants younger than 9 months. However, data from different parts of India suggest that 4-12% cases of measles occurred in infants younger than 9 months [8-10]. Analysis of outbreaks in urban slums in Delhi reported significant numbers of cases among infants younger than 9 months, with the youngest case being just one month old [11]. A study in Kerala (southern India), showed that among 43 children with measles, one-third acquired it prior to the age of vaccination [12]. During outbreaks, cases were identified among infants as young as 3-6 months in northern as well as southern India [13-15]. Likewise, an analysis of measles cases among infants younger than 1 year old, showed that 62.5% were younger than 9 months [16]. During an orphanage outbreak in Pune, five of six cases were younger than nine months; four were younger than 4 months old [17].

This information has been incorporated into the revised text as follows:

Revised text:

Data from different parts of India suggest that 4-12% cases of measles occurred in infants younger than 9 months [8-10]. Analysis of outbreaks in urban slums in Delhi reported significant numbers of cases among infants younger than 9 months, with the youngest case being just one month old [11]. A study in Kerala (southern India), showed that among 43 children with measles, one-third acquired it prior to the age of vaccination [12]. During outbreaks, cases were identified among infants as young as 3-6 months in northern as well as southern India [13-15]. Likewise, an analysis of measles cases among infants younger than 1 year old, showed that 62.5% were younger than 9 months [16]. During an orphanage outbreak in Pune, five of six cases were younger than nine months; four were younger than 4 months old [17].

Comment

Reviewer #3: There is the need for the authors to show how they arrived at the sample size in a more explicit manner. The relevance of the essence of collecting data on religion and cast has not been optimally utilised to arrive at a conclusion on whether these have any influence on the results obtained. There is the need to explain more about this in the discussion section. These could be related to the attitude and practice of the people towards vaccination or be responsible for susceptibility to infection.

Response: Thank you for the advice. 

Sample size calculation

This has been explained in more detail. Please see the revised text.

Relevance of collecting data on caste and religion

In India (and in many developing countries), the socio-demographic factors affecting infant vaccination rates, susceptibility to infectious diseases and access to healthcare (in general) include: socio-economic status, maternal education, caste, religion, and markers of woman empowerment (maternal age, parity). We have previously shown that the odds of seroconversion following measles vaccination was 0.78 (95% CI: 0.62, 0.98) among infants from Scheduled Castes/ Scheduled Tribes compared to the ‘Others’ caste group [Wagner AL, Mathew JL, Carlson BF, Kachoria AG, Bharti B, Suri V, Boulton ML. Measles Vaccination Immunogenicity and Association with Caste in Chandigarh, India. Am J Trop Med Hyg. 2022 Oct 3;107(5):1129-1131.]. 

However, in this analysis, there was no statistically significant impact of family religion or caste on the infant antibody levels, hence we have not emphasized it. 

Revised text:

We collected data on several additional variables that impact infant vaccination and susceptibility to disease, but none showed statistically significant impacts on anti-measles antibody levels.

Was there any difference in the data obtained in rural, urban and resettlement colonies of Chandigarh? This was not reflected in any way in the discussion. What was the relevance of the anthropomorphic measurements?

Response: 

We did not examine the data by the type of residence, hence are unable to assess its impact of any. We will conduct this analysis and report it in a separate publication.

In developing countries, malnourished infants generally fare worse in terms of vaccine-induced immunity and infectious disease burden. Therefore,, these data are routinely collected in research studies. We plan to undertake additional analysis to study the impact of nutritional status (if any). 

6. PLOS authors have the option to publish the peer review history of their article (what does this mean?). If published, this will include your full peer review and any attached files.

Do you want your identity to be public for this peer review? For information about this choice, including consent withdrawal, please see our Privacy Policy.

Reviewer #1: No

Reviewer #2: No

Reviewer #3: No

---

## [Decision Letter · Decision Letter 1]

15 Mar 2023

PONE-D-22-30962R1Dynamics of maternally transmitted anti-measles antibodies, and susceptibility to disease among infants in Chandigarh, India:  A prospective birth cohort study.PLOS ONE

Dear Dr. Mathew,

Thank you for submitting your manuscript to PLOS ONE. After careful consideration, we feel that it has merit but does not fully meet PLOS ONE’s publication criteria as it currently stands. Therefore, we invite you to submit a revised version of the manuscript that addresses the points raised during the review process.

We look forward to receiving your revised manuscript.

Kind regards,

Ghada Abdrabo Abdellatif Elshaarawy, M.D

Academic Editor

PLOS ONE

Journal Requirements:

Additional Editor Comments:

Make sure you thoroughly address the reviewers’ comments and concerns and ensure the manuscript is free of any editorial or grammatical errors.

Reviewers' comments:

Reviewer's Responses to Questions

**Comments to the Author**

1. If the authors have adequately addressed your comments raised in a previous round of review and you feel that this manuscript is now acceptable for publication, you may indicate that here to bypass the “Comments to the Author” section, enter your conflict of interest statement in the “Confidential to Editor” section, and submit your "Accept" recommendation.

Reviewer #1: (No Response)

Reviewer #3: All comments have been addressed

Reviewer #4: All comments have been addressed

Reviewer #5: (No Response)

2. Is the manuscript technically sound, and do the data support the conclusions?

Reviewer #1: Yes

Reviewer #3: Yes

Reviewer #4: Yes

Reviewer #5: Partly

3. Has the statistical analysis been performed appropriately and rigorously? 

Reviewer #1: Yes

Reviewer #3: Yes

Reviewer #4: Yes

Reviewer #5: No

4. Have the authors made all data underlying the findings in their manuscript fully available?

Reviewer #1: No

Reviewer #3: Yes

Reviewer #4: Yes

Reviewer #5: Yes

5. Is the manuscript presented in an intelligible fashion and written in standard English?

Reviewer #1: Yes

Reviewer #3: Yes

Reviewer #4: Yes

Reviewer #5: Yes

6. Review Comments to the Author

Reviewer #1: Review report

Tile of the manuscript: Dynamics of maternally transmitted anti-measles antibodies, and susceptibility to disease among infants in Chandigarh, India: A prospective birth cohort study.

Manuscript Number: PONE-D-22-30962R1.

Comments addressed: Almost all.

Revision Number: II

Review Comments

1. General Comments

Did the term ‘dynamics’ refers to the change in the antibody titer on both the mother and newborn, or only on newborn? Or only mothers? Is that an appropriate terminology?

What is the ICC among the measurements and why you used yours [auto-regressive among the five]?

2. Specific Comments

• You have explained that you are not clear with “The selection of the women and the newborns should be revisited. E.g., If mothers give town newborn, how did you manage?”. Hence, let me elaborate if a moth gave birth to two newborns, have you included both of them as a birth cohort or not?

• Laboratory SOP (WHO) should be stated as per the other references.

• I think it is better to avoid religion from the ‘methods’ section.

• What compensation was given to the study participants? What are the contents of the consent? Do you have consent for publication? What was done for those with extremely low antibody titer infants? Would you attach the ethical protocol? Since they are premature did maternal consent for drawing blood sample supported by your national regulations and laws?

• How did you assure the validity and reliability of the tests/measurements?

• Revisit all section for their standard scientific component.

• Try to revisit again the statistics, language and grammar.

Regards,

Reviewer #3: All concerns raised at the initial review had been addressed and the manuscript is currently presented in an acceptable standard for publication. This will add to be body of knowledge about the level of protection of maternal antibodies and the susceptibility of infants to measles before nine months.

Reviewer #4: (No Response)

Reviewer #5: General remarks

- The title: what do mean by dynamics? The study merely studies the attenuation of measles IG antibody titre. The susceptibility of the disease or infection?

- The data sets: the authors provided two separate data sets one for the mothers and other for infants, in the current format alignment into mother-infant data is impossible to re-use???

Specific comments:

- Line 121-123: more details about the random selection process for selecting the 30 anganwadis representing urban, rural and slums?

- Line 132: how many mothers (pregnant women) were approached at the outset of the study?

- Sample size calculation is not clear, based on personal judgment not considering other critical variables (exposure- risk/protective)???

- Why religion was included?

Statistical analysis:

The statistical analysis plan needs more clarification with the sequential steps used in data analysis- Bivariate analysis, auto-regressive covariance-log-linear regression (multivariate) and adjustment (weight) or sensitivity analysis is not clear.

- Bivariate analysis was done to 202 examine the UN-adjusted relationship between predictor variables and the outcome of interest: 203 anti-measles IgG antibody status and how it changes as a function of time. No bivariate analysis was displayed in the manuscript.

- Auto-regressive predictive model should be carried out

- How many variables were found significant out of the auto-regressive Covariance model?

- Name the possible predictors entered in your model, were any of them log-transformed according to their binary or multi-nominal nature?

- Use step-wise models to eliminate independent variables with possible multi-collinearity.

- No enough details about the sensitivity analysis and adjustment used in this study, and not displayed in the text or the tables.(details about loss to follow up? Or attrition should be included)

- Tables 1: replace descriptive statistics with bivariate analysis.

- Table 2: collapse your categories (income, religion, caste, parity, education, etc.,). Mention the details accompanied each model (significance).

7. PLOS authors have the option to publish the peer review history of their article (what does this mean?). If published, this will include your full peer review and any attached files.

Reviewer #1: No

Reviewer #3: No

Reviewer #4: **Yes: **Mohammad Ashraful Amin

Reviewer #5: **Yes: **Tarek Tawfik Amin

---

## [Author Response · Author response to Decision Letter 1]

11 Apr 2023

Response to Editorial and Reviewers’ comments

We are grateful to the Editor and the Reviewers for their careful appraisal of our manuscript. We have considered each comment carefully and addressed them all. 

Please find enclosed herewith:

1) A detailed point by point response to the comments and suggestions offered.

2) Revised manuscript with Track changes. All newly added text is also highlighted in blue color for easy identification. 

3) Revised manuscript (clean copy). 

We hope that the revised manuscript is acceptable to be considered for publication.

Thanking you,

Yours sincerely,

Joseph L. Mathew

26 March 2023

Journal Requirements:

Authors’ response

We have accessed the above links and ensured that the manuscript complies with the style requirements. In case we have missed anything, please advise us. 

Additional Editor Comments:

Make sure you thoroughly address the reviewers’ comments and concerns and ensure the manuscript is free of any editorial or grammatical errors.

Authors’ response

We have addressed all the comments and concerns in detail, and made extensive changes to the manuscript, as advised. 

Reviewer's Responses to Questions

Comments to the Author

1. If the authors have adequately addressed your comments raised in a previous round of review and you feel that this manuscript is now acceptable for publication, you may indicate that here to bypass the “Comments to the Author” section, enter your conflict of interest statement in the “Confidential to Editor” section, and submit your "Accept" recommendation.

Reviewer #1: (No Response)

Reviewer #3: All comments have been addressed

Reviewer #4: All comments have been addressed

Reviewer #5: (No Response)

Authors’ response

We have addressed all the specific comments and concerns in detail, and made extensive changes to the manuscript, as advised. Please see the details below.

2. Is the manuscript technically sound, and do the data support the conclusions?

Reviewer #1: Yes

Reviewer #3: Yes

Reviewer #4: Yes

Reviewer #5: Partly

Authors’ response

We have addressed all the specific comments and concerns in detail, and made extensive changes to the manuscript, as advised. Please see the details below.

3. Has the statistical analysis been performed appropriately and rigorously?

Reviewer #1: Yes

Reviewer #3: Yes

Reviewer #4: Yes

Reviewer #5: No

Authors’ response

We have addressed all the specific comments and concerns in detail, and made extensive changes to the manuscript, as advised. Please see the details below.

4. Have the authors made all data underlying the findings in their manuscript fully available?

Reviewer #1: No

Authors’ response

We have now provided the link to the online repository where all the raw data are available. Please see: https://doi.org/10.6084/m9.figshare.22190215.v2

Reviewer #3: Yes

Reviewer #4: Yes

Reviewer #5: Yes

5. Is the manuscript presented in an intelligible fashion and written in standard English?

Reviewer #1: Yes

Reviewer #3: Yes

Reviewer #4: Yes

Reviewer #5: Yes

6. Review Comments to the Author

Reviewer #1: Review report

Tile of the manuscript: Dynamics of maternally transmitted anti-measles antibodies, and susceptibility to disease among infants in Chandigarh, India: A prospective birth cohort study.

Manuscript Number: PONE-D-22-30962R1.

Comments addressed: Almost all.

Revision Number: II

Review Comments

Comment

1. General Comments

Did the term ‘dynamics’ refers to the change in the antibody titer on both the mother and newborn, or only on newborn? Or only mothers? Is that an appropriate terminology?

What is the ICC among the measurements and why you used yours [auto-regressive among the five]?

Authors’ response: 

Thank you for pointing out the confusion created by using the word “dynamics”. In this research study, we examined the change in titers of antibodies among the infants. Therefore, for simplicity, we have removed the word “dynamics” from the manuscript.

Revised text;

The word ‘dynamics’ has been deleted throughout the manuscript.

Authors’ response: 

The ICC is -0.029, but the negative number is due to variability within individuals exceeding variability across individuals at the same month. Given declines in antibody titers across time, this is expected.

We used autoregressive to account for changes over time, with the thought that variability would increase the more distance in months there are between observations. We added this sentence: “An autoregressive covariance structure was used to account for correlation between an infant’s visits, with increasing variability across farther apart months.”

Revised text:

An autoregressive covariance structure was used to account for correlation between an infant’s visits, with increasing variability across farther apart months.

2. Specific Comments

Comment

• You have explained that you are not clear with “The selection of the women and the newborns should be revisited. E.g., If mothers give town newborn, how did you manage?”. Hence, let me elaborate if a moth gave birth to two newborns, have you included both of them as a birth cohort or not?

Authors’ response: 

Thank you for the explanation. In this study, there were seven women who gave birth to twins. We included both pairs of twins in all seven cases. However, the sample is too small to conduct specific sub-analyses. 

Revised text:

In this study, there were seven women who gave birth to twins. We included both pairs of twins in all seven cases. However, the sample is too small to conduct specific sub-analyses

Comment

• Laboratory SOP (WHO) should be stated as per the other references.

Authors’ response: 

Thank you for this suggestion. The following two references have been added at No. 39 and 40.

Revised text:

39. Government of India. GUIDELINES FOR ACCREDITATION OF CHEMICAL AND

BIO-PESTICIDE TESTING LABORATORIES AS PER ISO 17025. 2017. https://ppqs.gov.in/sites/default/files/nabl-requirement.pdf (accessed October 22, 2018).

40. World Health Organization. Manual for the laboratory diagnosis of measles and rubella virus infection, 2nd ed. 2007. https://www.who.int/publications/i/item/WHO-IVB-07.01 (accessed November 12, 2018).

Comment

• I think it is better to avoid religion from the ‘methods’ section.

Authors’ response: 

Thank you for the comment. However, we did not understand whether the suggestion is to avoid religion in the ‘methods section’ alone, or the entire manuscript. We offer responses to both as follows: 

A) In India (and some other developing countries), childhood immunization coverage, access to healthcare, utilization of resources for healthcare, and epidemiology of disease as well as health indicators are often affected by religion. This may represent current or past (i.e. long entrenched) socio-demographic issues. Therefore, even though it is a sensitive subject, this data is generally collected in epidemiology surveys. 

B) In the methods section, the word ‘religion’ appears in a limited number of places viz. (i) Description of the Case Record Form mentioning that ‘religion’ was one of the socio-demographic characteristics; and (ii) Description of outcome measures. It is important to mention these to accurately describe the study procedures.

We have attempted to clarify the purpose of obtaining data on religion through the revised text below.

Revised text:

We also explored patterns by sociodemographic status (caste, religion, income, education) under a social determinants of health framework in order to ascertain whether any groups could be more affected by future measles outbreaks. 

Comment

• What compensation was given to the study participants? What are the contents of the consent? Do you have consent for publication? What was done for those with extremely low antibody titer infants? Would you attach the ethical protocol? Since they are premature did maternal consent for drawing blood sample supported by your national regulations and laws?

Authors’ response: 

(A) As per the institutional and national rules, no financial or monetary compensation was given to the study participants. 

(B) As per the principles of ethical biomedical research, and rules of the participating institutions, and in compliance with the laws of the land, participants were provided the following two documents: (i) a Participant Information Sheet, and (ii) an Informed Consent form. Both documents were made available in three languages (English, Hindi, and Punjabi). Separate documents were available for enrolment of children under the age of 18 years. The English version of both sets of documents for adult participants as well as infants, is included for examination (Annexures I and II).

Briefly, the Participant Information Sheet comprised details of the Investigators, funding agency, outline of the study, expected duration of participation, procedures followed during this study, description of the risks and discomforts to participants, benefits expected from the research, alternatives available to participants, confidentiality of the data/records of participants (which included a statement that the results would be published but not include any information that would identify participants, and would also include only grouped information from everyone in the study, and not individual information), treatment schedule(s), compensation and/or treatment(s) available to the participants in the event of a trial-related injury, whom to contact for trial related queries and the rights of participants in the event of any injury, the absence of financial compensation to take part in the study, responsibilities during participation in the study, a clear statement that participation is voluntary, that participants can withdraw from the study at any time and that refusal to participate will not involve any penalty or loss of benefits to which participants are otherwise entitled, a statement that the participants or their representatives would be notified in a timely manner if significant new findings develop during the course of the research which may their willingness to continue participation, a statement of foreseeable circumstances under which the participation may be terminated by the Investigator without the participant's consent, absence of additional costs to participants due participation in the study, consequences of a participant's decision to withdraw from the research and procedures for orderly termination of participation, the approximate number of participants to be enrolled in the study, and contact details of the Principal Investigator, Head of the Department, and the Ethics Committee/IRB of both institutions. 

The Informed Consent form included five additional statements to be signed individually, followed by a summary statement to be signed separately. The five statements included a confirmation that the participant has read and understood the information sheet dated 14 May 2015 for the study and had the opportunity to ask questions; (ii) a confirmation that the participant understands that participation in the study is voluntary and that he/she is free to withdraw at any time, without giving any reason, without their medical care or legal rights being affected, (iii) a statement that the participant understands that the Principal Investigator (PI) of the study, others working on the PI’s behalf, the Ethics Committee and the regulatory authorities will not need the participant’s permission to look at his/her health records both in respect of the current study and any further research that may be conducted in relation to it, even if the participant withdraws from the study, (iv) a statement that the participant understands that his/her identity will not be revealed in any information released to third parties or published. ,and (v) a confirmation that the participant agrees not to restrict the use of any data or results that arise from this study, provided such a use is only for scientific purpose(s).

(C) As described above, each participant signed the above, granting consent for publication of group data without revealing individual identity. 

(D) In this study, specimens were not processed in real-time, but serum samples were deep frozen and processed in batches. Therefore, the results of all infants were not available immediately, at all time points. Further, the national regulations do not permit infants to be vaccinated at an earlier age even if they are proved to be susceptible, except in the situation of an outbreak (wherein infants may be vaccinated as early as six months of age, but not earlier). Therefore, we did not do anything for infants with low antibody titres. However, all infants were followed up telephonically and requested to report in the event of any symptoms or signs suggestive of measles in them, or family members. 

(E) Please see Annexure I and Annexure II for the ethical protocols used in this study.

(F) As per our national laws and institutional ethics committee, any sampling of children (i.e. younger than 18 years of age) requires the consent of a parent or legal guardian, whether they were premature or not. This was followed in this study also. 

Comment

• How did you assure the validity and reliability of the tests/measurements?

Authors’ response: 

• As explained in the Methods section, all laboratory testing was performed in accordance with the current guidelines for performing such tests. The references for this have been cited at No. 39 and 40. This statement is already mentioned in the manuscript. 

• In addition, the manufacturer’s instructions given in the ELISA kits and online were followed meticulously. This statement is already mentioned in the manuscript. 

• Third, the Department of Virology of our institution (whose Faculty members are co-investigators in this study), is the Regional Virus Research and Diagnostic Laboratory (VRDL) of our country, with facilities and expertise for performing such tests. 

• Fourth, the Department of Virology of our institution is also a national Measles Laboratory of our country and is a WHO accredited site for measles and rubella testing. 

• Fifth, a single laboratory technician trained at the WHO accredited laboratory facility was dedicated for this study and performed all the tests. This negated the risk of inter-observer variation. This statement is already mentioned in the manuscript. 

• Sixth, we meticulously repeated tests in 20% of the samples selected at random, to ensure consistency. This statement is already mentioned in the manuscript. 

• Seventh, all samples with extremes of titres, and outlier values were re-tested twice to confirm accuracy of the results. This statement is already mentioned in the manuscript. 

We believe that these refinements confirm the accuracy and reliability of the laboratory results. However, all these details have not been included in the manuscript. Please advise If these should be stated therein.

Comment

• Revisit all section for their standard scientific component.

Authors’ response: 

We rechecked all sections of the manuscript for the scientific component and made a few changes for grater clarity. These are highlighted in the revised manuscript. 

Comment

• Try to revisit again the statistics, language and grammar.

Authors’ response:

Thank you for this valuable suggestion. We reviewed the paper for statistics, language, and grammar, and made several minor changes to improve the readability. These are highlighted in the revised manuscript. 

Comment

Reviewer #3: All concerns raised at the initial review had been addressed and the manuscript is currently presented in an acceptable standard for publication. This will add to be body of knowledge about the level of protection of maternal antibodies and the susceptibility of infants to measles before nine months.

Authors’ response:

Thank you for this kind response. We greatly appreciate this. 

Comment

Reviewer #4: (No Response)

Comment

Reviewer #5: General remarks

- The title: what do mean by dynamics? The study merely studies the attenuation of measles IG antibody titre. The susceptibility of the disease or infection?

Authors’ response: 

Thank you for pointing out the confusion created by using the word “dynamics”. In this research study, we examined the change in titers of antibodies among the infants. Therefore, for simplicity, we have removed the word “dynamics” from the manuscript, 

Revised text;

The word ‘dynamics’ has been deleted throughout the manuscript.

Comment

- The data sets: the authors provided two separate data sets one for the mothers and other for infants, in the current format alignment into mother-infant data is impossible to re-use???

Authors’ response: 

Thank you for pointing this out. Our data are available in two formats. A csv file that could not be uploaded, as well as an online repository. The link to the online dataset is as follows:

https://doi.org/10.6084/m9.figshare.22190215.v2

This makes available all the data used in this study.

Comment

Specific comments:

- Line 121-123: more details about the random selection process for selecting the 30 anganwadis representing urban, rural and slums?

Authors’ response: 

Chandigarh’s total population as per the decadal census data of 2011 was 1,055,450, with 11.3% of these being children. The population is distributed through three categories of residential areas viz. rural, urban and resettlement colonies (previously referred to as urban slums). 

Chandigarh has approximately 500 Anganwadi units which administer the Integrated Child Development Services (ICDS) scheme of the Government of India. These 500 Anganwadis are distributed across Chandigarh and are intended to serve a population of about 2000 people each. The Anaganwadis are therefore located across rural areas, urban areas, and resettlement colonies, in proportion to the population residing in the respective areas. Prior to initiating the study, the 500 Anganwadis were listed, and two meetings were held with the Administrators of the Anganwadis. Thereafter, a total of 30 Anganwadis were selected through a random selection process, and 15 back-up Anganwadis were selected as reserve. The 30 selected Anganwadis and 15 reserve Anganwadis were selected in proportion to the population residing in rural areas, urban areas, and resettlement colonies. This ensured that the population in this study represented the population of Chandigarh.

Revised text:

This population-based study was conducted during 2016-17, in Chandigarh, a city and Union Territory (UT) of India, that serves as the capital of two northern Indian states- Punjab and Haryana. Its population of over 1 million residents [24] resides in villages, urban areas, and resettlement colonies (previously referred to as “urban slums”). Chandigarh has 510 geographic units referred to as ‘Anganwadis’ [25], which administer the Governmental Integrated Child Development Services Scheme (ICDS), to address childhood malnutrition, infant vaccination, and antenatal registration of pregnant women. Each Anganwadi serves a population of about 2000 people. They are located across rural areas, urban areas, and resettlement colonies of Chandigarh. Prior to initiating the study, the 510 Anganwadis were listed, and a total of 30 were selected through a random selection process, and 15 back-up Anganwadis were selected as reserve. The 30 selected Anganwadis and 15 reserve Anganwadis were selected in proportion to the population residing in rural areas, urban areas, and resettlement colonies. This ensured that the population in this study represented the population of Chandigarh. 

Comment

- Line 132: how many mothers (pregnant women) were approached at the outset of the study?

Authors’ response: 

A total of 1483 pregnant women were approached at the outset of the study.

Revised text:

A total of 1483 pregnant women residing within the selected Anganwadis were approached during the third trimester, to explain (in the local vernacular) about the study, and its procedures.

Comment

- Sample size calculation is not clear, based on personal judgment not considering other critical variables (exposure- risk/protective)???

Authors’ response:

This study was designed to identify the decline in antibody titers with increasing age of infants. A previous pilot study conducted in our institution (cited as Reference 21) showed that the proportion of susceptible (i.e. unprotected) infants was 0%, 11.5%, 72%, and 94% at birth, 3 months, 6 months and 9 months of age, respectively [21]. Based on this data, we wanted to choose an adequate sample size to describe changes in antibody titers across the nine months from 94% to 72% to 12% to 0%, and significantly distinguish these proportions. Among various sample sizes tried out, a sample size of 200 provided that level of precision, at an alpha of 0.05 and a power of 80%. A sample size of 200 provided a margin of error of 4.6% for seropositivity values >90% and <10%, and 6.6% for values around 70%. This margin of error is adequate for our aim. Assuming approximately 50% attrition among infants over the duration of 9 months, we sought to enroll 400 mother-infant dyads. Since the exploration of antibody titers across sociodemographic groups was exploratory, we did not conduct separate sample size calculations for this.

Revised text:

This study was designed to identify the decline in antibody titers with increasing age of infants. A previous pilot study conducted in our institution showed that the proportion of susceptible (i.e. unprotected) infants was 0%, 11.5%, 72%, and 94% at birth, 3 months, 6 months and 9 months of age, respectively [21]. Based on this data, we wanted to choose an adequate sample size to describe changes in antibody titers across the nine months from 94% to 72% to 12% to 0%, and significantly distinguish these proportions. A sample size of 200 provided that level of precision, at an alpha of 0.05 and a power of 80%. It provided a margin of error of 4.6% for seropositivity values >90% and <10%, and 6.6% for values around 70%. This margin of error is adequate for our aim. Assuming approximately 50% attrition among infants over the duration of 9 months, we sought to enroll 400 mother-infant dyads. Since the exploration of antibody titers across sociodemographic groups was exploratory, we did not conduct separate sample size calculations for this.

Comment

- Why religion was included?

Authors’ response: 

In India (and some other developing countries), childhood immunization coverage, access to healthcare, utilization of resources for healthcare, and epidemiology of disease as well as health indicators are often influenced by religion. This may represent current or past (i.e. long entrenched) socio-demographic issues. Therefore, even though it is a sensitive subject, this data is generally collected in all epidemiology surveys. 

We have attempted to clarify the purpose of obtaining data on religion through the revised text below.

Revised text:

We also explored patterns by sociodemographic status (caste, religion, income, education) under a ‘social determinants of health’ framework in order to ascertain whether any groups could be more affected by future measles outbreaks. 

Comment

Statistical analysis:

The statistical analysis plan needs more clarification with the sequential steps used in data analysis- Bivariate analysis, auto-regressive covariance-log-linear regression (multivariate) and adjustment (weight) or sensitivity analysis is not clear.

Authors’ response: 

Thank you for the suggestion.

To characterize the decline in maternal antibodies among infants, descriptive statistics were first calculated to illustrate distributions of potential predictors. In a bivariate analysis, the geometric mean titer of anti-measles IgG at birth is displayed across different sociodemographic categories. 

A multivariable log-linear model was constructed to estimate the association between infants’ age and antibody titer. This model also included sociodemographic variables as an exploratory analysis.

For an additional sensitivity analysis, we adjusted for the loss to follow up. We did a weighted analysis, where we calculated inverse probability of staying in the study based on the same set of covariates used in the final multivariable logistic regression model. 

Revised text:

We examined the proportion of infants who fell into the negative, borderline, and positive categories of anti-measles IgG antibody status at 0, 3, 6, and 9 months. The mean antibody titer was also calculated at each of these time points.

To characterize the decline in maternal antibodies among infants, descriptive statistics were first calculated to illustrate distributions of potential predictors. In a bivariate analysis, the geometric mean titer of anti-measles IgG at birth is displayed across different sociodemographic categories. 

A multivariable log-linear model was constructed to estimate the association between infants’ age and antibody titer. This model also included sociodemographic variables as an exploratory analysis. An autoregressive covariance structure was used to account for correlation between an infant’s visits, with increasing variability across farther apart months. In order to obtain exp(β) and 95% confidence intervals (CIs) beta coefficients and lower/upper limits were exponentiated. Predictor variables were also tested for collinearity to ensure absence of strong linear association. The model included the following predictors: infant’s age, sex, breastfeeding status, family’s income, religion, caste, mother’s age, parity, education, and mother’s disease/vaccination status. All predictors, except mother’s age, were entered as categorical. Mother’s age was continuous and not log transformed.

For an additional sensitivity analysis, we adjusted for the loss to follow up: the enrolled sample declined to 253 at 3 months, 204 at 6 months, and 181 at 9 months. We have described the possible reasons for the loss to follow up in a separate manuscript [38]. In our sensitivity analysis, we adjusted for the inverse probability of infants staying in the study. We calculated these weights based on the same set of covariates used in the final multivariable logistic regression model (infant’s age, sex, breastfeeding status, family’s income, religion, caste, mother’s age, parity, education, and mother’s disease/vaccination status).

Comment

- Bivariate analysis was done to 202 examine the UN-adjusted relationship between predictor variables and the outcome of interest: 203 anti-measles IgG antibody status and how it changes as a function of time. No bivariate analysis was displayed in the manuscript.

Authors’ response:

Thank you for pointing this out. The data are shown in revised Table 1. 

Revised text:

Table 1. Baseline Demographic Characteristics of 413 infant-mother dyads at the time of birth. 

Characteristic Number Percentage Anti-measles IgG at birth

 Geometric mean titer 95% CI

Infant Sex 

Female 211 51.1 67 58 - 77

Male 202 48.9 62 53 - 71

Family’s monthly income 

< Rs 10,000 per month 186 45.0 67 58 - 77

Rs 10,000- 25,000 per month 121 29.3 64 53 - 78

Rs 25,000-50,000 per month 82 19.9 63 50 - 79

> Rs 50,000 per month 24 5.8 50 28 - 89

Family religion 

Hindu 340 82.3 66 59 - 74

Sikh 55 13.3 54 40 - 73

Muslim/Christian 18 4.4 67 43 - 104

Family Caste 

General 274 66.3 64 56 - 72

Other Backward Caste 54 13.1 66 49 - 90

SC/ST 85 20.6 65 52 - 81

Mother's Age (Mean, SD) 27.6 (4.6) 

18-24 years 105 25.4 63 50 - 78

25-29 years 181 43.8 61 52 - 72

≥30 years 127 30.8 70 60 - 83

Parity 

0 264 63.9 64 56 - 73

1 118 28.6 63 53 - 75

>2 31 7.5 69 52 - 91

Mother's Education 

None 16 3.9 75 44 - 127

Primary (<5 years of schooling) 16 3.9 77 47 - 127

 Middle (<8 years of schooling) 34 8.2 58 41 - 83

Higher secondary (<12 years of schooling) 115 27.9 64 53 - 76

College/University or higher 232 56.2 64 56 - 74

Mother's Measles Disease/Vaccination Status 

No vaccination, no disease 217 52.5 67 58 - 76

Unknown/missing for both 140 33.9 58 49 - 70

Vaccination, no disease 36 8.7 57 37 - 89

Disease, regardless of vaccination 20 4.8 108 76 - 152

Comment

- Auto-regressive predictive model should be carried out

Authors’ response: 

Thank you for this suggestion. However, this type of model is beyond the scope of this paper and so we respectfully decline. 

Comment

- How many variables were found significant out of the auto-regressive Covariance model?

Authors’ response: 

Two variables viz. infant age and mother’s age were found significant (as already indicated in Table 2). 

Comment

- Name the possible predictors entered in your model, were any of them log-transformed according to their binary or multi-nominal nature?

Authors’ response: 

Our model included the following predictors: infant’s age, sex, breastfeeding status, family’s income, religion, caste, mother’s age, parity, education, and mother’s disease/vaccination status. All predictors, except mother’s age, were entered as categorical. Mother’s age was continuous and not log transformed, because that reflected our a priori understanding that older ages would have gradually increased risk of history of infection.

Revised text:

The model included the following predictors: infant’s age, sex, breastfeeding status, family’s income, religion, caste, mother’s age, parity, education, and mother’s disease/vaccination status. All predictors, except mother’s age, were entered as categorical. Mother’s age was continuous and not log transformed. 

Comment

- Use step-wise models to eliminate independent variables with possible multi-collinearity.

Authors’ response: 

Thank you for the suggestion, However, stepwise models are becoming less popular and are in fact deprecated in current research (for instance, please see Smith J Big Data (2018) 5:32. https://doi.org/10.1186/s40537-018-0143-6), hence we did not do this, and respectfully decline to do it. 

Comment

- No enough details about the sensitivity analysis and adjustment used in this study, and not displayed in the text or the tables.(details about loss to follow up? Or attrition should be included)

Authors’ response: 

Thank you for pointing this out. We have addressed this through the following revised text.

Revised text:

For an additional sensitivity analysis, we adjusted for the loss to follow up: the enrolled sample declined to 253 at 3 months, 204 at 6 months, and 181 at 9 months. We have described the possible reasons for the loss to follow up in a separate manuscript [38]. In our sensitivity analysis, we adjusted for the inverse probability of infants staying in the study. We calculated these weights based on the same set of covariates used in the final multivariable logistic regression model (infant’s age, sex, breastfeeding status, family’s income, religion, caste, mother’s age, parity, education, and mother’s disease/vaccination status).

Comment

- Tables 1: replace descriptive statistics with bivariate analysis.

Authors’ response: 

We have revised Table 1 to show geometric mean titers with 95% CI as a bivariate analysis. 

Revised text:

Table 1. Baseline Demographic Characteristics of 413 infant-mother dyads at the time of birth. 

Characteristic Number Percentage Anti-measles IgG at birth

 Geometric mean titer 95% CI

Infant Sex 

Female 211 51.1 67 58 - 77

Male 202 48.9 62 53 - 71

Family’s monthly income 

< Rs 10,000 per month 186 45.0 67 58 - 77

Rs 10,000- 25,000 per month 121 29.3 64 53 - 78

Rs 25,000-50,000 per month 82 19.9 63 50 - 79

> Rs 50,000 per month 24 5.8 50 28 - 89

Family religion 

Hindu 340 82.3 66 59 - 74

Sikh 55 13.3 54 40 - 73

Muslim/Christian 18 4.4 67 43 - 104

Family Caste 

General 274 66.3 64 56 - 72

Other Backward Caste 54 13.1 66 49 - 90

SC/ST 85 20.6 65 52 - 81

Mother's Age (Mean, SD) 27.6 (4.6) 

18-24 years 105 25.4 63 50 - 78

25-29 years 181 43.8 61 52 - 72

≥30 years 127 30.8 70 60 - 83

Parity 

0 264 63.9 64 56 - 73

1 118 28.6 63 53 - 75

>2 31 7.5 69 52 - 91

Mother's Education 

None 16 3.9 75 44 - 127

Primary (<5 years of schooling) 16 3.9 77 47 - 127

 Middle (<8 years of schooling) 34 8.2 58 41 - 83

Higher secondary (<12 years of schooling) 115 27.9 64 53 - 76

College/University or higher 232 56.2 64 56 - 74

Mother's Measles Disease/Vaccination Status 

No vaccination, no disease 217 52.5 67 58 - 76

Unknown/missing for both 140 33.9 58 49 - 70

Vaccination, no disease 36 8.7 57 37 - 89

Disease, regardless of vaccination 20 4.8 108 76 - 152

Comment

- Table 2: collapse your categories (income, religion, caste, parity, education, etc.,). Mention the details accompanied each model (significance).

Authors’ response: 

Thank you for the suggestion. We collapsed parity, religion, and caste. Some of the categories in these variables had low cell counts. However, we retained the categories with adequate cell counts in income and maternal education.

We redid the analysis accounting for this change, but the numbers in Table 2 only changed minimally and did not affect the results.

Revised text:

Parity greater than 2 was combined into one category. For religion, we collapsed together the Muslim and Christian categories; and for caste, the Scheduled Caste (SC) and Scheduled Tribe (ST) categories, due to low cell counts.

 

Annexure I: Documents pertaining to adult participants (mothers)

Participant Information Sheet and Consent to Participate in a Research Study

PRINICPAL INVESTIGATORS: Dr. Joseph L. Mathew

Department of Pediatrics, Advanced Pediatrics Centre, PGIMER Matthew L. Boulton, MD, MPH 

Department of Epidemiology,

University of Michigan 

CO-INVESTIGATORS: Prof. Sourabh Dutta

Department of Pediatrics

PGIMER Dr. Vikas Suri

Department of Internal Medicine, PGIMER

 Prof. Vanita Suri 

Department of Gynecology and Obstetrics, PGIMER Prof. R K Ratho

Department of Virology, 

PGIMER 

 Dr. Bhavneet Bharti

Department of Pediatrics, 

PGIMER Dr. Mini P Singh

Department of Virology, 

PGIMER

SPONSOR: Trehan Foundation and The University of Michigan School of Public Health

You are invited to be a part of a research study with approximately 1500 participants that is planned to identify people who are at risk of getting measles. Measles is a disease that makes people have a red rash and fever. Measles can make people very sick and/or have serious complications. Getting measles disease or getting vaccinated against measles protects people from getting measles in the future. We want to learn more about people in Chandigarh and whether or not they are protected from measles. The information in this document is meant to help you decide if you want to participate in this study. Please feel free to ask if you have any queries or concerns.

1. What is your expected duration of participation?

Your participation in this study will last for a total of 15-25 minutes. 

2. What procedures will be followed during this study?

If you agree to participate in this research study, you will be asked some questions related to your age, gender, education, occupation, and prior exposure to measles disease or measles vaccination. This will be done by a person trained to conduct this interview and is expected to last for 5-15 minutes. You will be asked to let us measure your height and weight and to permit collection of a small amount of blood. This will be done by cleaning your arm with a chemical (70% isopropyl alcohol), followed by pricking a vein to draw some blood. This should take about 5 minutes of your time. The blood sample will be used to measure your level of protection against measles (measles IgG antibody level). 

3. What are the risks and discomforts to you?

We do not expect any discomfort or risk to you during the interview process. However, collection of the blood sample can cause temporary discomfort and pain during the procedure. There is also the theoretical but unlikely possibility of introducing infection and/or causing blood loss in a person at risk of bleeding/clotting disorders. However, these risks are expected to be small since trained professionals will collect the blood. The procedure will be performed as per standard guidelines using all precautions to minimize risk.

4. What benefits are expected from this research?

 You will have the direct benefit of knowing the results of your blood test, which we can use to determine if you are at risk for measles. This will help you to decide whether you need to be vaccinated to protect yourself against measles. Additionally, others may indirectly benefit from this research because we will be able to estimate the risk of measles infection among the people of Chandigarh. We may also be able to develop more effective strategies to reduce the number of measles cases in Chandigarh and throughout India.

5. What are the alternatives available to you?

You have the complete freedom to choose to participate or not participate in this study. Even if you decide to participate now, you may change your mind and stop at any time. Since there are no treatments planned in this study, there are no alternative treatments.

6. Are the data/records of the Participant kept confidential?

We plan to publish the results of this study, but will not include any information that would identify you. Information that is published will be presented as grouped information from everyone who participates in our study, not just your information. 

All your data and records will be kept confidential. They will be stored securely at Post Graduate Institute of Medical Education and Research. They will not be disclosed to any personnel who do not require access to the same. They will be disclosed on request only to the Institutional Ethics Committee and/or External/Internal Research Monitoring Board.

7. What will be the treatment schedule(s)?

This is not an interventional research study and no treatment will be provided during this study.

8. What compensation and/or treatment(s) are available to the Participant in the event of a trial-related injury?

This study is not a trial, and hence, there is no possibility of trial-related injury. 

9. Whom to contact for trial related queries and what are the rights of Participants in the event of any injury? 

This research study is not a trial. Nevertheless, you are free to contact the Principal Investigators at any time during the study. The contact details are given at the end of this sheet. In case you are not satisfied with this, you are free to contact: The Head of Department, Department of Pediatrics, PGIMER, Chandigarh 160012. In addition, you are also free to contact: The Convenor, Institute Ethics Committee, PGIMER Chandigarh 160012. You are also free to contact the University of Michigan Health Sciences and Behavioral Sciences Institutional Review Board, 2800 Plymouth Road, Bldg. 520, Suite 1169, Ann Arbor, MI 48109-2800, (734) 936-0933 [or toll free, (866) 936-0933], [for international calls, 001+734-936-0933], irbhsbs@umich.edu.

10. Are the participants paid to take part in this study?

There will be no payment or any other financial incentive for participation in this study. However, expenses related to transportation can be reimbursed on submission of appropriate documents. If the Participant is found to be at risk of developing measles and choose to be vaccinated for protection, the vaccine will be provided free of cost if the Participant is not eligible for the Government funded vaccine. Additionally, we will provide blood type testing to the Participant if requested. 

11. What are your responsibilities during participation in the study?

If you choose to participate in this study, you would be expected to:

• Provide answers to the questions that are asked by the interviewer. These are related to information about your age, gender, education, occupation, household income, type of residence (urban or rural), and previous history of measles disease or measles vaccination. 

• Permit measurement of your height and weight.

• Permit collection of a blood sample for laboratory testing (measurement of measles IgG antibody level). 

12. Statement that participation is voluntary, that the Participant can withdraw from the study at any time and that refusal to participate will not involve any penalty or loss of benefits to which the Participant is otherwise entitled.

Your participation in this study is purely voluntary. Each participant will have the right to withdraw from the study at any time without having to give reasons for doing so. Refusal to participate will not involve any penalty or loss of benefits to which you are otherwise entitled. If you decide to withdraw your participation, we may still be able to use information provided by you. However, if you desire that no information provided by you should be used, we will not use it.

13. Statement that the Participant or Participant's representative will be notified in a timely manner if significant new findings develop during the course of the research which may affect the Participant's willingness to continue participation will be provided.

You will be notified in a timely manner if significant new findings develop during the course of the research study which may affect your willingness to continue participation in this study.

14. Statement of foreseeable circumstances under which the Participant's participation may be terminated by the Investigator without the Participant's consent. 

We do not foresee any circumstances wherein your participation may be terminated without your consent.

15. Additional costs to the Participant that may result from participation in the study.

No additional costs are expected to you on account of your participation in the study. 

16. The consequences of a Participant's decision to withdraw from the research and procedures for orderly termination of participation by Participant.

If you decide to terminate your participation in this study but desire to receive the data available until the time of termination of your participation, the available data will be provided to you.

17. A statement that the particular treatment or procedure may involve risks to the Participant (or to the embryo or fetus, if the Participant is or may become pregnant), which are currently unforeseeable.

Not applicable

18. Approximate number of Participants enrolled in the study.

A total of 1500 participants are expected to be enrolled in this study.

19. Any other pertinent information.

Not applicable

Contact persons: 

For further information / questions, you can contact:

Principal Investigator: Dr. Joseph L. Mathew, Department of Pediatrics, PGIMER, Chandigarh 160012.

Tel: 0172-2755357, Email: mathew.joseph@pgimer.edu.in, Fax: 0172-2744401

You are also free to contact: The Head of Department, Department of Pediatrics, PGIMER, Chandigarh 160012

In case of conflicts, you can contact the Convener of the Institutional Ethics Committee at the following address:

Prof. S. Malhotra, Convener, Institutional Ethics Committee, Department of Pharmacology, PGIMER, Chandigarh

You are also free to contact:

University of Michigan Health Sciences and Behavioral Sciences Institutional Review Board, 2800 Plymouth Road, Bldg. 520, Suite 1169, Ann Arbor, MI 48109-2800, (734) 936-0933 [or toll free, (866) 936-0933], [for international calls, 001+734-936-0933], Email: irbhsbs@umich.edu.

Consent to Participate

 Participant's initial 

1. I confirm that I have read and understood the information sheet dated 14 May 2015 for the above study and have had the opportunity to ask questions. 

2. I understand that my participation in the study is voluntary and that I am free to withdraw at any time, without giving any reason, without my medical care or legal rights being affected. 

3. I understand that the Principal Investigator (PI) of the study, others working on the PI’s behalf, the Ethics Committee and the regulatory authorities will not need my permission to look at my health records both in respect of the current study and any further research that may be conducted in relation to it, even if I withdraw from the study. I agree to this access. However, I understand that my identity will not be revealed in any information released to third parties or published. 

4. I agree not to restrict the use of any data or results that arise from this study, provided such a use is only for scientific purpose(s) 

5. I hereby give my full, free and voluntary, informed consent to participate in the above-mentioned research study. 

Consent

By signing this document, I am agreeing to participate in the study. Even so, I may change my mind and withdraw my participation from the study at any time. I have been given a copy of this document for my records and one copy will be kept with the study records. All my questions about the study have been answered and I understand what I am being asked to do. I may contact the researchers if I think of any question later.

Signature (or thumb impression) of the Participant or parent/legally acceptable representative if the Participant is below 18 years of age 

Name of signatory 

Date 

If the Participant was not able to read the informed consent information by himself/herself, a witness is necessary.

I, the witness, have been present at all times during the consent process. Every question asked by the Participant was answered and the Participant agrees to participate in this research voluntarily.

Signature of the Witness 

Name of signatory 

Date 

 

I certify that the nature and purpose, the potential benefits, and possible risks associated with participating in this research have been explained to the participant or participants’ parent/ legally acceptable representative if the participant is below 18 years of age.

Signature of the research study personnel

Name of signatory 

Date 

Signature of the investigator

Name of signatory 

Date 

 

Annexure II: Documents pertaining to infant participants

Participant Information Sheet and Consent to Participate in a Research Study

PRINICPAL INVESTIGATORS: Dr. Joseph L. Mathew

Department of Pediatrics, Advanced Pediatrics Centre, Post Graduate Institute of Medical Education and Research Matthew L. Boulton, MD, MPH 

Department of Epidemiology,

University of Michigan 

CO-INVESTIGATORS: Prof. Sourabh Dutta

Department of Pediatrics

PGIMER Dr. Vikas Suri

Department of Internal Medicine, PGIMER

 Prof. Vanita Suri

Department of Gynecology and Obstetrics, PGIMER Prof. R K Ratho

Department of Virology,

PGIMER

 Dr. Bhavneet Bharti

Department of Pediatrics,

PGIMER Dr. Mini P Singh

Department of Virology,

PGIMER

SPONSOR: Trehan Foundation and The University of Michigan School of Public Health

You are invited to include your infant to be a part of a research study, with approximately 200 infant participants (1500 total participants), that is planned to identify people who are at risk of getting measles. Measles is a disease that makes people have a red rash and fever. Measles can make people very sick and/or have serious complications. Getting measles disease or getting vaccinated against measles protects people from getting measles in the future. We want to learn more about people in Chandigarh and whether or not they are protected from measles. The information in this document is meant to help you decide if you want to participate in this study. Please feel free to ask if you have any queries or concerns.

1. What is your expected duration of participation?

Your infant’s participation in this study will be for a period of 12 months starting from birth. A total of five contacts are planned at birth, 3months, 6 months, 9 months and 12 months. Each contact will be for a duration of 15-25 minutes. 

2. What procedures will be followed during this study?

If you agree for your infant to participate in this research study, you will be asked some questions related to your infant’s date of birth, birth weight, gestational age, and exposure to measles disease or measles vaccination. This will be done by a person trained to conduct this interview and is expected to last for 5-15 minutes. You will be asked to let us measure the height and weight and to permit collection of a small amount of blood from your infant. This will be done by cleaning his/her arm with a chemical (70% isopropyl alcohol), followed by pricking a vein to draw some blood. This should take about 5-10 minutes. The blood sample will be used to measure your level of protection against measles (measles IgG antibody level). 

3. What are the risks and discomforts to you?

We do not expect any discomfort or risk to you or your infant during the interview process. However, collection of the blood sample can cause temporary discomfort and pain to the infant during the procedure. There is also the theoretical but unlikely possibility of introducing infection and/or causing blood loss in a person at risk of bleeding/clotting disorders. However, these risks are expected to be small since trained professionals will collect the blood. The procedure will be performed as per standard guidelines using all precautions to minimize risk.

4. What benefits are expected from this research?

You will have the direct benefit of knowing the results of your infant’s blood test, which we can use to determine if your infant is at risk for measles. This will help you to decide whether you need your infant to be vaccinated to protect him/her against measles. Additionally, others may indirectly benefit from this research because we will be able to estimate the risk of measles infection among the people of Chandigarh. We may also be able to develop more effective strategies to reduce the number of measles cases in Chandigarh and throughout India.

5. What are the alternatives available to you?

You have the complete freedom to choose whether your infant should participate or not participate in this study. Even if you decide to participate now, you may change your mind and withdraw your infant’s participation at any time. Since there are no treatments planned in this study, there are no alternative treatments.

6. Are the data/records of the Participant kept confidential?

We plan to publish the results of this study, but will not include any information that would identify you. Information that is published will be presented as grouped information from everyone who participates in our study, not just your information. 

All data and records of your infant will be kept confidential. They will be stored securely at Post Graduate Institute of Medical Education and Research. They will not be disclosed to any personnel who do not require access to the same. They will be disclosed on request only to the Institutional Ethics Committee and/or External/Internal Research Monitoring Board

7. What will be the treatment schedule(s)?

This is not an interventional research study and no treatment will be provided during this study.

8. What compensation and/or treatment(s) are available to the Participant in the event of a trial-related injury?

This study is not a trial, and hence, there is no possibility of trial-related injury. 

9. Whom to contact for trial related queries and what are the rights of Participants in the event of any injury? 

This research study is not a trial. Nevertheless, you are free to contact the Principal Investigators at any time during the study. The contact details are given at the end of this sheet. In case you are not satisfied with this, you are free to contact: The Head of Department, Department of Pediatrics, PGIMER, Chandigarh 160012. In addition, you are also free to contact: The Convenor, Institute Ethics Committee, PGIMER Chandigarh 160012. You are also free to contact the the University of Michigan Health Sciences and Behavioral Sciences Institutional Review Board, 2800 Plymouth Road, Bldg. 520, Suite 1169, Ann Arbor, MI 48109-2800, (734) 936-0933 [or toll free, (866) 936-0933], [for international calls, 001+734-936-0933], irbhsbs@umich.edu.

10. Are the participants paid to take part in this study?

There will be no payment or any other financial incentive for your infant’s participation in this study. However, expenses related to transportation can be reimbursed on submission of appropriate documents. If your infant is old enough and is found to be at risk of developing measles and you choose to vaccinate him/her for protection, the vaccine will be provided free of cost. Additionally, we will provide blood type testing to the infant participant if requested. 

11. What are your responsibilities during participation in the study?

If you choose to let your infant participate in this study, you would be expected to:

• Provide answers to the questions that are asked by the interviewer. These are related to obtain information about your infant’s date of birth, birth weight, gestational age, gender, history of exposure to measles disease or measles vaccination, household income, type of residence (urban or rural) and infant feeding practices. 

• Permit measurement of your infant’s height and weight.

• Permit collection of a blood sample for laboratory testing (measurement of measles IgG antibody level). 

• Permit the above mentioned interview and blood sample collection at four additional visits when the infant’s age is 3 months, 6 months, 9 months and 12 months. These visits are designed to coincide with visits for routine vaccination and/or physical examination. 

12. Statement that participation is voluntary, that the Participant can withdraw from the study at any time and that refusal to participate will not involve any penalty or loss of benefits to which the Participant is otherwise entitled.

Your infant’s participation in this study is purely voluntary. Each enrolled infant will have the right to drop out of the study at any time without having to give reasons for doing so. They will continue to receive the same standard of care available to all such infants. Refusal to participate will not involve any penalty or loss of benefits to which your infant is otherwise entitled. If you decide to withdraw your infant’s participation, we may still be able to use information provided by you. However, if you desire that no information provided by you should be used, we will not use it.

13. Statement that the Participant or Participant's representative will be notified in a timely manner if significant new findings develop during the course of the research which may affect the Participant's willingness to continue participation will be provided.

You will be notified in a timely manner if significant new findings develop during the course of the research study which may affect your willingness to continue participation of your infant in this study.

14. Statement of foreseeable circumstances under which the Participant's participation may be terminated by the Investigator without the Participant's consent. 

In case you are unable or unwilling to follow the schedule required in this study, the Principal Investigator may have to terminate your infant’s participation in the study, even without your consent. 

15. Additional costs to the Participant that may result from participation in the study.

No additional costs are expected to you on account of your infant’s participation in the study. 

16. The consequences of a Participant's decision to withdraw from the research and procedures for orderly termination of participation by Participant.

If you decide to terminate the participation of your child in this study but desire to receive the data available till the time of termination of your infant’s participation, the available data will be provided to you.

17. A statement that the particular treatment or procedure may involve risks to the Participant (or to the embryo or fetus, if the Participant is or may become pregnant), which are currently unforeseeable.

Not applicable

18. Approximate number of Participants enrolled in the study.

A total of 200 infants are expected to be enrolled in this study, which will have 1500 total participants of various ages.

19. Any other pertinent information.

Not applicable

Contact persons: 

For further information / questions, you can contact:

Principal Investigator: Dr. Joseph L. Mathew, Department of Pediatrics, PGIMER, Chandigarh 160012.

Tel: 0172-2755357, Email: mathew.joseph@pgimer.edu.in, Fax: 0172-2744401

You are also free to contact: The Head of Department, Department of Pediatrics, PGIMER, Chandigarh 160012

In case of conflicts, you can contact the Convener of the Institutional Ethics Committee at the following address:

Prof. S. Malhotra, Convener, Institutional Ethics Committee, Department of Pharmacology, PGIMER, Chandigarh

You are also free to contact:

University of Michigan Health Sciences and Behavioral Sciences Institutional Review Board, 2800 Plymouth Road, Bldg. 520, Suite 1169, Ann Arbor, MI 48109-2800, (734) 936-0933 [or toll free, (866) 936-0933], [for international calls, 001+734-936-0933], Email: irbhsbs@umich.edu.

Consent to Participate

 Participant's parent/ guardian initial 

1. I confirm that I have read and understood the information sheet dated 14 May 2015 for the above study and have had the opportunity to ask questions. 

2. I understand that my infant’s participation in the study is voluntary and that I am free to withdraw at any time, without giving any reason, without my infant’s medical care or legal rights being affected. 

3. I understand that the Principal Investigator (PI) of the study, others working on the PI’s behalf, the Ethics Committee and the regulatory authorities will not need my permission to look at my infant’s health records both in respect of the current study and any further research that may be conducted in relation to it, even if I withdraw from the study. I agree to this access. However, I understand that my infant’s identity will not be revealed in any information released to third parties or published. 

4. I agree not to restrict the use of any data or results that arise from this study, provided such a use is only for scientific purpose(s) 

5. I hereby give my full, free and voluntary, informed consent for my infant to participate in the above-mentioned research study. 

Consent

By signing this document, I am agreeing for my infant to participate in the study. Even so, I may change my mind and withdraw my infant from the study at any time. I have been given a copy of this document for my records and one copy will be kept with the study records. All my questions about the study have been answered and I understand what I am being asked to do. I may contact the researchers if I think of any question later.

Signature (or thumb impression) of the parent/legally acceptable representative 

Name of signatory 

Date 

If the Participant’s parent/guardian was not able to read the informed consent information by himself/herself, a witness is necessary.

I, the witness, have been present at all times during the consent process. Every question asked by the Participant’s parent/guardian was answered and the Participant’s parent/guardian agrees to participate in this research voluntarily.

Signature of the Witness 

Name of signatory 

Date 

I certify that the nature and purpose, the potential benefits, and possible risks associated with participating in this research have been explained to the participants’ parent/ legally acceptable representative as the participant is below 18 years of age.

Signature of the research study personnel

Name of signatory 

Date 

Signature of the investigator

Name of signatory 

Date

---

## [Decision Letter · Decision Letter 2]

15 May 2023

PONE-D-22-30962R2Maternally transmitted anti-measles antibodies, and susceptibility to disease among infants in Chandigarh, India:  A prospective birth cohort study.PLOS ONE

Dear Dr. Mathew,

Thank you for submitting your manuscript to PLOS ONE. After careful consideration, we feel that it has merit but does not fully meet PLOS ONE’s publication criteria as it currently stands. Therefore, we invite you to submit a revised version of the manuscript that addresses the points raised during the review process.

We look forward to receiving your revised manuscript.

Kind regards,

Ghada Abdrabo Abdellatif Elshaarawy, M.D

Academic Editor

PLOS ONE

Journal Requirements:

Additional Editor Comments:

Make sure you thoroughly address the reviewers’ comments and concerns and ensure the manuscript is free of any editorial or grammatical errors.

Reviewers' comments:

Reviewer's Responses to Questions

**Comments to the Author**

1. If the authors have adequately addressed your comments raised in a previous round of review and you feel that this manuscript is now acceptable for publication, you may indicate that here to bypass the “Comments to the Author” section, enter your conflict of interest statement in the “Confidential to Editor” section, and submit your "Accept" recommendation.

Reviewer #3: All comments have been addressed

Reviewer #4: All comments have been addressed

Reviewer #5: All comments have been addressed

2. Is the manuscript technically sound, and do the data support the conclusions?

Reviewer #3: Yes

Reviewer #4: Yes

Reviewer #5: Yes

3. Has the statistical analysis been performed appropriately and rigorously? 

Reviewer #3: Yes

Reviewer #4: Yes

Reviewer #5: Yes

4. Have the authors made all data underlying the findings in their manuscript fully available?

Reviewer #3: Yes

Reviewer #4: Yes

Reviewer #5: Yes

5. Is the manuscript presented in an intelligible fashion and written in standard English?

Reviewer #3: Yes

Reviewer #4: Yes

Reviewer #5: Yes

6. Review Comments to the Author

Reviewer #3: The authors should do a few English corrections especially where words were omitted within a sentence eg line 135. The authors also need to adopt a uniform writing style either American or British eg the spelling of 'titer' or 'titre'.

Reviewer #4: (No Response)

Reviewer #5: Substantial improvement of the manuscript, applying the appropriate data analysis and interpretation and solid conclusion of the research work.

Only the figure needs some enhancement (colors scheme) and layout. the time should be explicitly mentioned (0, 3. 6, 9 months)

7. PLOS authors have the option to publish the peer review history of their article (what does this mean?). If published, this will include your full peer review and any attached files.

Reviewer #3: **Yes: **Dr Babatunde Adewale

Reviewer #4: **Yes: **Mohammad Ashraful Amin

Reviewer #5: **Yes: **Tarek Tawfik Amin

---

## [Author Response · Author response to Decision Letter 2]

29 May 2023

PONE-D-22-30962R2

Maternally transmitted anti-measles antibodies, and susceptibility to disease among infants in Chandigarh, India: A prospective birth cohort study.

Response to Editorial and Reviewers’ comments

We are grateful to the Editor and the Reviewers for their careful re-appraisal of our manuscript. We have glad that the reviewers have accepted the revised manuscript, and that two minor revisions have been suggested. We have considered both comments carefully and addressed them. 

Please find enclosed herewith:

1) A detailed point by point response to the comments and suggestions offered.

2) Revised manuscript with Track changes. All newly added text is also highlighted in blue color for easy identification. 

3) Manuscript (revised manuscript clean copy). 

We hope that the revised manuscript is acceptable to be considered for publication.

Thanking you,

Yours sincerely,

Joseph L. Mathew

29 May 2023

PLOS ONE

Dear Dr. Mathew,

Thank you for submitting your manuscript to PLOS ONE. After careful consideration, we feel that it has merit but does not fully meet PLOS ONE’s publication criteria as it currently stands. Therefore, we invite you to submit a revised version of the manuscript that addresses the points raised during the review process.

Authors’ response: We have followed the instructions given above, and enclosed the three files as specified. 

Authors’ response: There is no change to the financial disclosure statement. 

Authors’ response: Thank you for the suggestion. Standard laboratory methods and protocols were used. As there is no novel method used, we have cited the relevant references in the manuscript.

We look forward to receiving your revised manuscript.

Kind regards,

Ghada Abdrabo Abdellatif Elshaarawy, M.D

Academic Editor

PLOS ONE

Journal Requirements:

Authors’ response: We have followed the instructions given above. 

Additional Editor Comments:

Make sure you thoroughly address the reviewers’ comments and concerns and ensure the manuscript is free of any editorial or grammatical errors.

Authors’ response: We have followed the instructions given above, and done our best to ensure that the revised manuscript is free from errors. 

Reviewers' comments:

Reviewer's Responses to Questions

Comments to the Author

1. If the authors have adequately addressed your comments raised in a previous round of review and you feel that this manuscript is now acceptable for publication, you may indicate that here to bypass the “Comments to the Author” section, enter your conflict of interest statement in the “Confidential to Editor” section, and submit your "Accept" recommendation.

Reviewer #3: All comments have been addressed

Reviewer #4: All comments have been addressed

Reviewer #5: All comments have been addressed

Authors’ response: Thank you for confirming that all comments have been addressed.

2. Is the manuscript technically sound, and do the data support the conclusions?

Reviewer #3: Yes

Reviewer #4: Yes

Reviewer #5: Yes

Authors’ response: Thank you for the confirmation. 

3. Has the statistical analysis been performed appropriately and rigorously? 

Reviewer #3: Yes

Reviewer #4: Yes

Reviewer #5: Yes

Authors’ response: Thank you for the confirmation. 

4. Have the authors made all data underlying the findings in their manuscript fully available?

Reviewer #3: Yes

Reviewer #4: Yes

Reviewer #5: Yes

Authors’ response: Thank you for the confirmation. 

5. Is the manuscript presented in an intelligible fashion and written in standard English?

Reviewer #3: Yes

Reviewer #4: Yes

Reviewer #5: Yes

Authors’ response: Thank you for the confirmation. 

6. Review Comments to the Author

Reviewer #3: The authors should do a few English corrections especially where words were omitted within a sentence eg line 135. The authors also need to adopt a uniform writing style either American or British eg the spelling of 'titer' or 'titre'.

Authors’ response: Thank you for pointing out the error. At all places, the word is spelt as “titer”.

Revised text:

In Line 135, the sentence has been changed as shown below:

Original sentence: Those were willing to participate…….

Revised sentence: Those who were willing to participate………

Reviewer #4: (No Response)

Reviewer #5: Substantial improvement of the manuscript, applying the appropriate data analysis and interpretation and solid conclusion of the research work.

Only the figure needs some enhancement (colors scheme) and layout. the time should be explicitly mentioned (0, 3. 6, 9 months)

Authors’ response: 

Thank you for the positive comments. 

As advised, the figure has been substantially modified with the following changes:

1. We did not fully understand the comment about the colors scheme. Therefore, we have uploaded the Figure in greyscale version as well as color version. The Editor in Chief is requested to choose whichever is better in appearance.

2. The time (0,3,6,9 months) has been explicitly mentioned.

3. The legend “Infant age in months (mo)” has been incorporated just beneath the ages.

4. The layout has been modified for greater clarity. 

5. The Y axis on the right side has been re-formatted as per the scale of the Geometric Mean Titre (GMT).

6. The Y axis label “Geometric Mean Titre (GMT [IU/ml])” has been placed right next to the Y axis.

7. PLOS authors have the option to publish the peer review history of their article (what does this mean?). If published, this will include your full peer review and any attached files.

Do you want your identity to be public for this peer review? For information about this choice, including consent withdrawal, please see our Privacy Policy.

Reviewer #3: Yes: Dr Babatunde Adewale

Reviewer #4: Yes: Mohammad Ashraful Amin

Reviewer #5: Yes: Tarek Tawfik Amin

Authors’ response: Thank you for the confirmation. 

The authors have no difficulty with the peer review history of the article being published.

---

## [Editor Report · Decision Letter 3]

30 May 2023

Maternally transmitted anti-measles antibodies, and susceptibility to disease among infants in Chandigarh, India:  A prospective birth cohort study.

PONE-D-22-30962R3

Dear Dr. Mathew,

We’re pleased to inform you that your manuscript has been judged scientifically suitable for publication and will be formally accepted for publication once it meets all outstanding technical requirements.

Kind regards,

Ghada Abdrabo Abdellatif Elshaarawy, M.D

Academic Editor

PLOS ONE
---

## [Editor Report · Acceptance letter]

4 Jun 2023

PONE-D-22-30962R3 

Maternally transmitted anti-measles antibodies, and susceptibility to disease among infants in Chandigarh, India:  A prospective birth cohort study. 

Dear Dr. Mathew:

I'm pleased to inform you that your manuscript has been deemed suitable for publication in PLOS ONE. Congratulations! Your manuscript is now with our production department. 

Kind regards, 

on behalf of

Dr. Ghada Abdrabo Abdellatif Elshaarawy 

Academic Editor

PLOS ONE